



# How relevant are frequency changes of weather regimes for understanding climate change signals in surface precipitation in the North Atlantic-European sector? – a conceptual analysis with CESM1 large ensemble simulations

Luise J. Fischer[1,2,a], David N. Bresch[2,3], Dominik Büeler[4,1], Christian M. Grams[4,3], Matthias Röthlisberger[1], and Heini Wernli[1]

[1]Institute for Atmospheric and Climate Science, ETH Zürich, Switzerland
[2]Institute for Environmental Decisions, ETH Zürich, Switzerland
[3]now at: Federal Office of Meteorology and Climatology, MeteoSwiss, Switzerland
[4]Institute of Meteorology and Climate Research (IMK-TRO), Department Troposphere Research, Karlsruhe Institute of Technology (KIT), Karlsruhe, Germany
[a]present address: South Pole, Melbourne, Australia

**Correspondence:** Luise J. Fischer (luise.fischer@usys.ethz.ch)

**Abstract.** Climate change affects the climatology of surface precipitation in spatially in-homogeneous ways and it is challenging to identify and quantify the contribution of atmospheric circulation changes to this pattern. Various methods have been developed to characterize the large-scale atmospheric circulation and assess its changes, e.g., by classifying the flow into so-called weather regimes or circulation types. Several studies have then related frequency changes of these regimes due to global warming to changes in surface weather parameters. However, even without regime frequency changes, the climatology of surface parameters may change due to so-called regime intensity changes (e.g., a particular regime becomes on average wetter or drier). In this study, the question of how relevant frequency changes of weather regimes are for understanding climate change signals in surface precipitation is addressed with a novel conceptual framework. For every regime $i$, a spatially varying parameter $\gamma_i(P)$ is introduced, which corresponds to the ratio of the contributions from regime frequency vs. regime intensity changes to the climate change signal of precipitation $P$. Conceptual considerations show that $\gamma_i(P)$ is (i) proportional to the relative change of regime frequency, (ii) proportional to the regime-specific anomaly of precipitation, and (iii) inversely proportional to the climate change effect on regime intensity. The combination of these independent and competing factors makes the study of $\gamma_i(P)$ interesting and insightful. As a specific example application of this framework, we consider a 7-category weather regime classification in the North Atlantic-European sector and large ensemble simulations with the CESM1 climate model under the RCP8.5 emission scenario for the periods 1990-1999 and 2091-2100. Considering $\gamma_i(P)$ for surface precipitation $P$ in this simulation setup reveals that (1) $\gamma$ values are typically less than 0.3 and therefore, to first order, frequency changes of WRs are of secondary importance for explaining climate change signals in $P$ – in contrast, the intensity changes dominate, which are to a large degree, but not entirely, related to the so-called thermodynamic effects of global warming; (2) the main reason for the generally low values of $\gamma$ are the comparatively small WR frequency changes and the limited regime-specific anomalies of $P$, in particular over continental Europe; and (3) $\gamma$ values tend to be slightly larger for precipitation variables that are less con-





strained by thermodynamic arguments, i.e., $\gamma$ for the number of wet days is larger than $\gamma$ for the number of heavy precipitation days. In summary, this study provides a generally applicable framework to quantify climate change effects of regime frequency changes on surface parameters, it illustrates the key conditions that must be fulfilled such that these frequency changes can become relevant, and, at least in our application, it shows that these conditions are generally not fulfilled.

# 1 Introduction

Given the complex chaotic nature of the large-scale atmospheric flow with the repeated occurrence of similar flow patterns, over many decades meteorologists and climatologists have attempted to sort the daily flow patterns into distinct categories. Early examples are the so-called "Grosswetterlagen" by Hess and Brezowsky (1952) with 30 types for Europe and the weather types by Schüepp (1959) for the European Alps with 6 classes and 33 types (Bader and Richner, 2018). Namias (1968) discussed the role of "weather typing" for extended and long-range forecasts of daily precipitation in the US and the history of these ideas that go back to the late 19th century. These early classifications were, as any classification of atmospheric flows, based on specific choices of the region of interest, the variables and vertical levels to be considered, and the number of categories to be determined.

While these early classification schemes were based on mainly subjective criteria, more objective methods, for instance based on principle component analysis (sometimes combined with clustering analysis), were developed in the last decades (e.g., Barnston and Livezey, 1987) and led to many flow classifications. Different names have been used for the flow categories, e.g., circulation types/regimes or weather types/regimes. The COST Action 733 compared 71 such circulation type classifications for Central Europe, which illustrates the multitude of existing approaches (Tveito and Huth, 2016). An important example is the classification by Vautard (1990) with four weather regimes (WRs) identified for the North Atlantic-European region in winter, based on low-pass-filtered geopotential height at 700 hPa (i.e., with the transient synoptic variability filtered out). This classification has been extended recently by Grams et al. (2017), identifying seven WRs in all seasons based on low-pass-filtered geopotential height at 500 hPa and in a similarly large domain. Addressing the fundamental question whether WRs represent physical modes of the atmosphere, or are merely useful statistical categorizations, Hochman et al. (2021) used concepts from dynamical systems theory and found clear evidence that most WRs identified by Grams et al. (2017) are physically meaningful and suitable for addressing circulation responses in a changing climate.

One of the key properties of these classic WRs is that they characteristically link the lower-frequency and thus more predictable evolution of the large-scale circulation to regional surface weather. They have thus become a popular tool in weather and climate science in recent years. In the context of today's climate, WRs are used to understand and classify the variability of regional surface weather and related socio-economic parameters in various regions around the world. For instance, they have been shown to modulate, in certain regions, the occurrence of extreme temperature, extreme precipitation, or surface



wind speed (Yiou and Nogaj, 2004; Pasquier et al., 2019; Mastrantonas et al., 2020; Coe et al., 2021; Zhang and Wang, 2021), renewable energy production (Jerez and Trigo, 2013; Grams et al., 2017; van der Wiel et al., 2019; Drücke et al., 2021), aerosol transport and concentration (Gaetani et al., 2021), air pollution (Maddison et al., 2021), or human excess mortality (Huang

et al., 2020). WRs are thus increasingly identified in operational weather forecasts – particularly on subseasonal-to-seasonal lead times, on which predictability arises from slowly varying planetary-scale phenomena that modulate WR variability on multi-daily to weekly timescales (Palmer, 1999; Lang et al., 2020). Categorical WR forecasts as predictors for regional surface weather can thus be beneficial at these lead times compared to direct model output of surface weather variables on a grid-point level (Vigaud et al., 2018; Bloomfield et al., 2021).

In the context of past and future climates, WRs are often used to understand the relative contributions of changes in large-scale variability (i.e., in WR occurrence) and of thermodynamic changes to surface weather changes between different climate conditions. For instance, Yiou et al. (2012) showed that pattern or frequency changes of the four main North Atlantic-European WRs are too small to explain the substantial temperature changes Europe experienced between the Medieval Warm Period (10th to 13th centuries) and the Little Ice Age (16th to 19th centuries). Likewise, Horton et al. (2015) argued that more recent trends in

Northern Hemisphere extreme surface temperature were primarily driven by thermodynamic changes, although changes in WR frequency and duration were important for specific regions. Furthermore, Huguenin et al. (2020) found that most WR frequency changes towards the end of the 21st century are small and not consistent across different climate models. Nevertheless, applying such a distinction of thermodynamic and dynamic climate change effects does not only improve our physical understanding but also helps in assessing how robust the projected regional surface weather changes might be based on their contributing

processes. For instance, future changes in the frequency and intensity of Northern Hemisphere cold extremes in winter will result from a complex superposition of – amongst others – changes in global mean temperature, in cold air outbreak intensity due to Arctic amplification, in the frequency of stratospheric polar vortex disruptions due to changes in stratospheric dynamics, and in midlatitude Rossby wave activity due to changes in baroclinicity. The last of these processes is directly related to WR occurrence. However, there is rather low consensus on future changes in WR occurrence and their role for surface weather

changes. For instance, for the positive and negative phases of the North Atlantic Oscillation (NAO), which are considered as distinct WRs in several North Atlantic/European WR classifications, Cattiaux et al. (2013) found an increasing frequency of the negative NAO phase in CMIP5 projections, while Fabiano et al. (2021) detected more positive phases of the NAO in CMIP6 projections. The relative contribution of these future WR frequency changes to surface weather changes has also been shown to differ, depending on the region, season, and surface parameter. While Riediger and Gratzki (2014) and Santos et al. (2016)

found, next to a likely thermodynamic background signal, a relatively important contribution of future WR changes to changes in surface temperature and precipitation in Western and Central Europe, Cattiaux et al. (2013) attributed a rather minor role to these changes.

This study aims at contributing to this research theme, which connects climate change, WRs, and surface weather, by introducing a conceptual framework that provides insight into why, where and when WR frequency changes can matter for

understanding climate change effects on surface weather. The conceptual framework will be illustrated with a particular choice of WRs in the North Atlantic-European region, climate simulations, and surface weather parameters. Importantly, we do not





attempt to provide a "final answer" regarding the role and relevance of WR frequency changes for understanding climate change signals, but rather develop a framework that allows quantitatively analysing the factors that determine this role. To be more specific, the main objectives of this study are to:

1. provide a conceptual view on the conditions that determine whether WR frequency changes are relevant for explaining climate change signals of a specific parameter $\phi$, in comparison to intensity changes. To quantitatively asses this relevance, we will introduce a field $\gamma_i(\phi)$, which can be calculated for any choice of WR classification with regimes $i = 1...n$ and parameter $\phi$.

      2. quantify the relevance of WR frequency changes for understanding effects of climate change specifically for different
95       aspects of precipitation $P$ in Europe. To this end, we will use output from large ensemble simulations with the climate model CESM1, and a specific 7-category WR classification. We will quantify $\gamma_i(\phi)$ for this specific choice of WRs and climate simulations, and investigate whether WR frequency changes are more important for understanding climate change effects on mean precipitation $\phi = P$, on the number of wet days $\phi = N_{wet}$, or on the number of heavy precipitation days $\phi = N_{heavy}$, respectively.

Section 2 first introduces the climate simulation data sets and the WR classification used in this study, and then, in Sect. 3, the field $\gamma_i(\phi)$, which serves to address the above objective 1. Example applications of this concept to climate change signals in surface precipitation are presented in Sect. 4 (objective 2), and in Sect. 5 the main conclusions are summarized and critically discussed.

## 2   Climate simulations and WR identification

This study uses global large ensemble climate simulations for a historic and future period under the RCP8.5 emission scenario (Sect. 2.1), and a 7-category all-season WR classification for the North Atlantic-European region is applied to output from these simulations (Sect. 2.2 and 2.3).

### 2.1   CESM1-LE climate simulations

The coupled ocean-atmosphere climate simulations used in this study were performed with version 1 of the Community Earth
System Model (CESM1) (Hurrell et al., 2013). Six-hourly fields of geopotential height at 500 hPa, $Z_{500}$, and surface precipitation $P$, on a regular grid with a horizontal resolution of approximately $1°$, were obtained through reruns of the CESM1-LE simulations (Kay et al., 2015), that are described in more detail in Röthlisberger et al. (2020). In this study, data from simulations with external forcing from two specific decades are investigated separately, one period covering the years 1990–1999 and the second one covering the years 2091–2100, henceforth referred to as historical and future (or end-of-century) simulations,
respectively. For both periods, the data sets consist of 35 ensemble members, each 10 years long, yielding 350 years with historical and future climate conditions, respectively. Note that due to the coupled nature of these simulations, the individual members only share the same external forcing but are independent in terms of their evolution of sea surface temperatures and



the phases of, for instance, the El Niño Southern Oscillation. Therefore, the 350 years in each climate period yield a very large sample of possible atmospheric responses to the external forcing that is specific for the respective decade. As this study will

focus on climate change effects on seasonal-mean $P$, model output was accumulated over the standard seasons December-February (DJF), March-May (MAM), June-August (JJA), and September-November (SON), and values of $P$ are given in units of $\mathrm{mm\,d^{-1}}$. In addition to $P$, we will also consider the seasonal number of wet days $N_{wet}$ (defined with a threshold of $1\,\mathrm{mm\,d^{-1}}$) and of heavy precipitation days $N_{heavy}$ (defined with a threshold of the 99th percentile of daily precipitation values in the respective season in the historical simulations).

### 125 2.2 Year-round North Atlantic-European WRs

The 7-category all-season North Atlantic-European WR definition by Grams et al. (2017) is used in this study. The main reason for this specific choice is that this classification can be applied to all days of the year, providing a year-round categorization of the large-scale flow pattern in the North Atlantic-European sector. Grams et al. (2017) identified the WRs in the region extending from 80°W to 40°E and 30°N to 90°N. They are based on an empirical orthogonal function (EOF) analysis of

six-hourly low-pass filtered anomaly fields of $Z_{500}$, taken from ERA-Interim reanalysis data (Dee et al., 2011) from 1979 to 2015, and subsequent k-means clustering in the EOF space yielding the 7 WRs (i.e., 7 clusters). Further details about this WR classification and how it can be applied to data from model simulations can be found in Grams et al. (2017) and Büeler et al. (2021), respectively.

The names of the WRs are based on the main flow pattern they represent and are as follows: Zonal Regime (ZO), Atlantic

Trough (AT), Atlantic Ridge (AR), Scandinavian Trough (ScTr), Greenland Blocking (GL), European Blocking (EuBL), and Scandinavian Blocking (ScBL). The seven WRs explicitly capture different flavors of (strong) zonal flows and the occurrence of atmospheric blocking over Greenland, Central Europe, and Scandinavia, respectively. If a $Z_{500}$ anomaly field does not meet the classification criteria for any of the seven WR patterns, it is classified into a "no-regime category" (no). The interested reader finds illustrations of the average sea level pressure field and near-surface wind speed anomalies over Europe associated

with these WRs in Grams et al. (2017, their Fig. 2). The no-regime is very close to the overall climatology. For the time period 1979-2015, Grams et al. (2017) reported averaged annual WR frequencies of 31.5% for the no-regime, and between 9.0% (AT) and 10.9% (ScBL) for the seven main WRs.

### 2.3 Identification of WRs in CESM1-LE

The WR classification introduced in Sect. 2.2 was applied to output from CESM1-LE, both for the historical and future

periods. Importantly, when attributing a daily $Z_{500}$ anomaly field from CESM1-LE to one of the WRs, we used the original ERA-Interim-based cluster mean $Z_{500}$ patterns of the WRs identified by Grams et al. (2017), i.e., no separate EOF analysis was performed with fields from CESM1-LE. This pragmatic approach has the advantage that WR patterns remain unaltered and climate change can only modify the frequency of these patterns [this is further justified by theoretical argumentation that climate change will primarily change the frequency rather than the structure of quasi-stationary regimes, see Palmer (1999)].



If separate EOF analyses were performed for the two climate simulation periods, then WRs would change both their pattern and frequency, making the interpretation of the results from the decomposition approach (Sect. 3) less straightforward.

In this paragraph we provide additional technical information about how the WR classification, originally developed for reanalysis data, was applied to climate model data. For more details, the reader is referred to Sect. 4.2.4 in Fischer (2021) and Sect. 2.2 in Büeler et al. (2021). The key step is the projection of a simulated daily $Z_{500}$ anomaly field to the seven $Z_{500}$ patterns

of the ERA-Interim WRs, which in essence corresponds to the spatial correlation of the two fields. The anomalies of $Z_{500}$ are computed relative to the climatology in the respective climate period. Following the approach by Michel and Rivière (2011), we then computed, for each daily $Z_{500}$ anomaly and for each regime $i = 1...7$, a non-dimensional regime index $I_i$. This index corresponds to normalized anomalies of the projection for each regime $i$ relative to the mean projection in the ERA-Interim period (1979-2015), and the normalization is done with the climatological standard deviation of the projection in the ERA-

Interim period (1979-2015). Eventually, to determine the active weather regime at a given time, a set of so-called life-cycle criteria is applied to the time series of the regime indices. A regime $i$ is considered "active" if its $I_i(t)$ is maximum among all seven indices at time $t$ and equal to or above a threshold value of 0.98 for at least five consecutive days. This threshold value differs very slightly from the value of 1.0 used by Grams et al. (2017) in their ERA-Interim study. We decided to modify the threshold value such that we obtain the same percentage of no-regime days in the CESM1-LE historic simulation as we find

in ERA-Interim for the years 1990-1999 (30.8% averaged over all seasons). The fact that with such a soft tuning, we could obtain the same projection rate to any of the 7 regimes (almost 70%) in CESM1-LE as in ERA-Interim, serves as a qualitative confirmation that the North Atlantic-European flow variability in the historic simulations compares favourably with reanalyses. For identifying WRs in the future climate simulations, the same procedure and the same threshold value of 0.98 are applied, leading to a slight reduction in the no-regime frequency (30.0%).

Figure 1 shows the WR frequencies in DJF and JJA in the historic and future climate simulations. Note that these values differ from the ones reported in Fischer (2021, their Table B.2), as those were found to be affected by a programming error. In DJF, the regimes AT, ZO, and ScTr are more frequent (about 11-15%) than the "blocked regimes" EuBL, ScBL, and GL (about 6-9%), whereas the opposite is true in JJA (3-8% vs. 11-18%). These seasonal differences are consistent with the reanalysis-based results in Grams et al. (2017). Note that we expect some quantitative differences of $f_{hist,i}$ in CESM1 compared to ERA-

Interim, as CESM1 data for the historic period is only representative of the external forcing in the 1990s but not necessarily for the modes of decadal variability in this period, see discussion in Sect. 2.1. Most of the WR frequency changes from historic to future climate conditions ($\Delta f_i = f_{eoc,i} - f_{hist,i}$) are modest. Relatively large values of $\Delta f_i$ occur in DJF for the regimes AT ($+1.7\%$, corresponding to a relative increase of $12.6\%$) and AR ($-1.2\%$, corresponding to a relative decrease of $12.0\%$), and in JJA for the no-regime ($-3.3\%$, corresponding to a relative decrease of $10.1\%$) and in particular for the regime AR ($+1.8\%$,

corresponding to a relative increase of $24.7\%$).




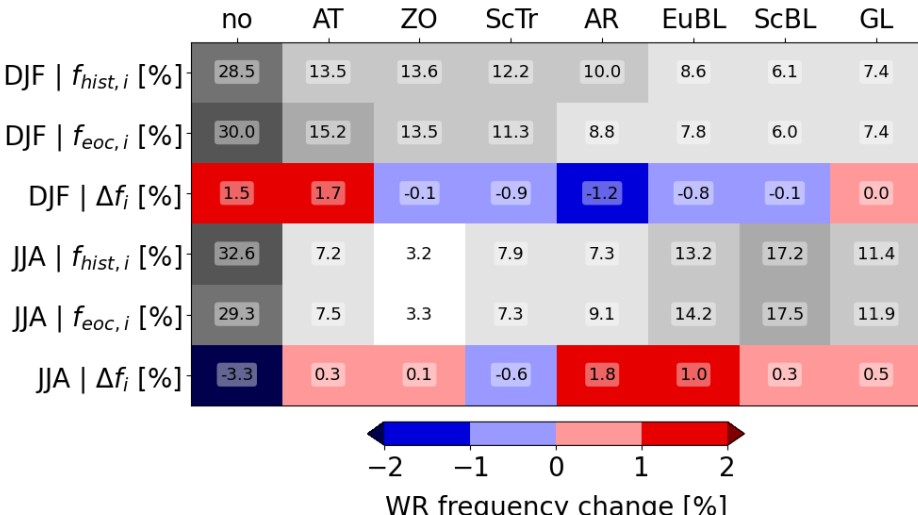

**Figure 1.** WR frequencies (in %) in DJF and JJA, respectively, in the CESM1 historic and future simulations ($f_{hist,i}$ and $f_{eoc,i}$), computed with 6-hourly $Z_{500}$ fields. Also listed are the values of the climate change effect on WR frequencies, $\Delta f_i$.

## 3  Quantifying WR-contributions to climate change signals

In this section, as the key methodological novelty of this study, we introduce a conceptual framework that serves quantifying the relevance of climate change effects on WR frequencies for understanding climate change signals in a surface weather variable $\phi$, such as precipitation. The starting point is to decompose this signal into contributions from changes related to WR frequency, $\Delta f_i$, and from changes related to what we call WR intensity, $\Delta \phi_i$, for instance the change in precipitation during WR $i$. This idea is written out explicitly in the following equations.

Let $\phi_p$ be the climatological value of the variable of interest (e.g., seasonal mean precipitation) in period $p \in \{hist, eoc\}$, and $\phi_{p,i}$ the climatological mean value of $\phi$ at all time steps in WR $i$ in period $p$. The climate change signal on $\phi$, i.e. the difference $\Delta \phi = \phi_{eoc} - \phi_{hist}$ can then be written as

$$\Delta \phi = \sum_{i=1}^{n} (f_{eoc,i} \phi_{eoc,i} - f_{hist,i} \phi_{hist,i}), \tag{1}$$

where $f_{p,i}$ is the frequency of WR $i$ in period $p$ and the summation is over all $n$ WRs of a certain WR classification. Introducing the notation $\Delta \phi_i = \phi_{eoc,i} - \phi_{hist,i}$ and $\Delta f_i = f_{eoc,i} - f_{hist,i}$ to denote the climate change effect on the WR-specific intensity $\phi_i$ and on the frequency of WR $i$, respectively, Eq. (1) can be re-written as

$$\Delta \phi = \sum_{i=1}^{n} \left[ (f_{hist,i} + \Delta f_i)(\phi_{hist,i} + \Delta \phi_i) - f_{hist,i} \phi_{hist,i} \right] \tag{2a}$$

$$= \sum_{i=1}^{n} \left[ \underbrace{f_{hist,i} \Delta \phi_i}_{(i)} + \underbrace{\Delta f_i \phi_{hist,i}}_{(ii)} + \underbrace{\Delta f_i \Delta \phi_i}_{(iii)} \right] \tag{2b}$$





Considering for instance precipitation, then the three terms in Eq. (2b) represent (i) the "contribution from the WR-specific precipitation intensity change to the climate change signal in precipitation", (ii) the "contribution from WR frequency change to the climate change signal in precipitation", and (iii) a "residual" term due to concurrent changes in WR-specific precipitation intensity and WR frequency. It is important to mention that such decompositions were used in several previous studies and most

likely the first time by Cassano et al. (2007), who quantified the role of frequency changes of synoptic flow patterns for global warming related changes in precipitation. Similar decompositions using circulation patterns or WRs were applied, for instance, by Horton et al. (2015) to temperature trends, and by Cattiaux et al. (2013) to biases in climate models. And last but not least, this climate change partitioning method can also be applied to a specific weather system (e.g., extratropical cyclones), by using a binary classification of days influenced by the weather system or not (Zappa et al., 2015). Several of the previous studies

used the terminology introduced by Cassano et al. (2007) and referred to terms (i) and (ii) as the thermodynamic and dynamic (or circulation) change components, respectively.

Here, we further decompose term (ii), by considering the deviation of the mean WR-specific pattern from the climatology: $\phi^*_{hist,i} = \phi_{hist,i} - \phi_{hist}$, thus $\phi^*_{hist,i}$ is the WR-specific anomaly. Note that $\sum_{i=1}^{n} \phi^*_{hist,i} = 0$ by design. The interpretation of, e.g. a positive value of $\phi^*_{hist,i}$ is that during an active WR $i$, the variable $\phi$ is on average larger than in the climatology. Clearly,

a WR-based analysis of a variable $\phi$ is particularly insightful if the values of $\phi^*_{hist,i}$ differ significantly from zero, at least for some $i$.

The second term on the right hand side of Eq. (2b) can thus be written as

$$\Delta f_i \phi_{hist,i} = \Delta f_i (\phi_{hist} + \phi^*_{hist,i}) \tag{3a}$$
$$= \Delta f_i \phi_{hist} + \Delta f_i \phi^*_{hist,i} \tag{3b}$$

Inserting this in Eq. (2b), the decomposition of the climate change signal on $\phi$ reads

$$\phi_{eoc} - \phi_{hist} = \sum_{i=1}^{n} \big[ \underbrace{f_{hist,i} \Delta \phi_i}_{(i)} + \underbrace{\Delta f_i \phi_{hist}}_{(iia)} + \underbrace{\Delta f_i \phi^*_{hist,i}}_{(iib)} + \underbrace{\Delta f_i \Delta \phi_i}_{(iii)} \big] \tag{4}$$

Terms (i) and (iii) in Eq. (4) are the same as in Eq. (2b), and the other two terms in the square brackets of Eq. (4) represent (iia) the "effect of the WR frequency change on the climate change signal in precipitation independent of the WR-specific deviation from the climatology", (iib) the "effect of the WR frequency change on the climate change signal in precipitation

due to the WR-specific deviation from the climatology".

There are two benefits of this additional decomposition: Firstly, the term (iib) can take either sign irrespective of the sign of $\Delta f_i$, which makes this term more easily interpretable than term (ii). For instance, if a particularly dry weather regime ($\phi^*_{hist,i} < 0$) becomes more frequent ($\Delta f_i > 0$) then term (iib) is negative, while term (ii) is always positive for a positive $\Delta f_i$. Secondly, the term (iia), i.e. $\Delta f_i \phi_{hist}$, which can be significantly larger than the other terms, cancels to zero when summed

over all WRs (because $\sum_{i=1}^{n} \Delta f_i = 0$). Making the assumption that the residual (term iii) is small (to be verified a posteriori),





Eq. (4) can be reduced to

$$\phi_{eoc} - \phi_{hist} \approx \sum_{i=1}^{n} \big[ \underbrace{f_{hist,i}\Delta\phi_i}_{(i)} + \underbrace{\Delta f_i \phi_{hist,i}^*}_{(iib)} \big] \tag{5}$$

The objective of this paper can now be formulated mathematically. It is to investigate the contribution of the frequency change term (iib) compared to the intensity change term (i), for each WR $i$. The modulus of this ratio is denoted as $\gamma_i(\phi)$ and given by

$$\gamma_i(\phi) := \left| \frac{\frac{\Delta f_i}{f_{hist,i}}}{\frac{\Delta\phi_i}{\phi_{hist,i}^*}} \right|. \tag{6}$$

Note that we consider the modulus of this ratio as we are interested in how the magnitudes of frequency and intensity changes compare, while the sign of the ratio would indicate whether frequency and intensity changes share the same sign or oppose one another. Furthermore, from Eq. 5 we can also define an overall $\gamma$, defined as

$$\gamma_{\text{overall}}(\phi) = \left| \frac{\sum_{i=1}^{n} \big[ \Delta f_i \phi_{hist,i}^* \big]}{\sum_{i=1}^{n} \big[ f_{hist,i}\Delta\phi_i \big]} \right| \tag{7}$$

We will consider $\gamma_{\text{overall}}$ at the end of the paper, but first focus on the WR-specific $\gamma_i(\phi)$.

Note that $\gamma_i(\phi)$ is a field and can be calculated at every model grid point. The question, whether frequency changes of WR $i$ are important for explaining climate change signals in $\phi$, can now be posed more precisely as "how large are the fields $\gamma_i(\phi)$", where $i = 1,....n$. A large $\gamma_i(\phi)$ indicates a greater importance of frequency changes of WR $i$ for explaining climate change signals in $\phi$ than a small $\gamma_i(\phi)$. Interestingly, $\gamma_i(\phi)$ depends on three independent factors (see Eq. 6):

1. $\frac{\Delta f_i}{f_{hist,i}}$, which is the relative frequency change of WR $i$ due to climate change. Since climate change does not completely alter the large-scale circulation, it is reasonable to assume that this factor has values on the order of 0.1, i.e., if a certain WR occurs with a frequency of 10% in the historical climate it will occur with a frequency in the range of 9-11% in the end-of-century climate. $\gamma_i$ is directly proportional to this factor.

2. $\Delta\phi_i$, which is the climate change effect on the intensity of the parameter $\phi$ in the WR $i$. It corresponds to the average value of $\phi$ on days in WR $i$ in the end-of-century climate, $\phi_{eoc,i}$, minus the same average in the historical climate, $\phi_{hist,i}$. $\gamma_i$ is inversely proportional to this factor, and therefore $\gamma_i$ becomes larger when this factor is small, i.e., if climate change does not affect the WR-specific intensity of parameter $\phi$. It is difficult to estimate the magnitude of this factor a priori. The expression for $\gamma_i$ tells us that we should compare the magnitude of this term with $\phi_{hist,i}^*$. Both this and the previous factor depend on the magnitude of climate change.

3. $\phi_{hist,i}^*$, which is the WR-specific anomaly of parameter $\phi$. This factor does not depend on climate change; it is rather a measure for the ability (or skill) of the WR classification to separate contrasting situations in terms of the parameter $\phi$. Two idealized cases shed further light on the role of this factor: If days were randomly attributed to one of the WRs,





then the average of $\phi$ would be the same for all WRs and therefore $\phi^*_{hist,i} = 0$ for all $i$. In this case $\gamma_i = 0$, i.e., WR frequency changes are completely irrelevant because the WRs do not distinguish different scenarios of $\phi$. The opposite case is a highly skillful WR classification, which separates days with strongly positive anomalies of $\phi$ from those with strongly negative anomalies of $\phi$. In this situation the magnitude of $\phi^*_{hist,i}$ can be large, i.e., comparable to the day-to-day variance of $\phi$. Now, what matters for $\gamma_i$ is the ratio of factors (2) and (3), $\frac{\Delta\phi_i}{\phi^*_{hist,i}}$. Note that this is not the ratio $\frac{\Delta\phi_i}{\phi_{hist,i}}$, i.e., the relative intensity change of $\phi$ in WR $i$, which would be much smaller. Rather it is the ratio of intensity change to the WR-specific *anomaly* of $\phi$. For a "very good WR classification", $\phi^*_{hist,i}$ might be 10 times larger than $\Delta\phi_i$, and in this case we would obtain $\gamma_i = 1$ (for the above-mentioned assumption that $\frac{\Delta f_i}{f_{hist,i}}$ is in the order of 0.1), i.e., in this case the frequency change of WR $i$ would be equally important as the intensity change of WR $i$. As an aside, we note that different measures were previously used for assessing the quality of a WR classification for a given parameter $\phi$, e.g., the Brier skill score (Schiemann and Frei, 2010) or the "coefficient of efficiency" (Madonna et al., 2021).

In summary, $\gamma_i(\phi)$ depends on the climate change effect on the frequency of the WR and on the WR-specific intensity change of $\phi$, as well as on the skill of the WR classification to separate different states of $\phi$. As we will see in the remainder of this paper, it appears difficult to obtain values of $\gamma_i(\phi)$ that are O(1). This is the case at least for the parameter we investigate (surface precipitation, $\phi = P$), for the climate simulations we use (CESM1-LE for the RCP8.5 scenario), and for our WR-classification in the North Atlantic-European region. We also show that $\gamma_i(P)$ varies spatially, mainly because of spatial variations of the third factor, i.e. the ability of the WR classification to represent local variability of the considered parameter $\phi$. Of course, we do not exclude the possibility that other WR classifications in other regions applied to other parameters could lead to larger values of $\gamma_i(\phi)$ than the ones documented in this study. This would require a combination of a more skillful classification, larger frequency changes, and smaller intensity changes.

For the visualization of $\gamma_i(P)$, we decided that it is meaningful to mask regions where the absolute climate change signal of $P$, i.e. $P_{eoc} - P_{hist}$, is below a certain threshold. In these regions where the climate change signal is weak, the question how much WR frequency changes contribute becomes obsolete and potentially large values of $\gamma_i(P)$ would be difficult to interpret. For each parameter and in each season, we have subjectively chosen a threshold such that $\gamma_i$ is masked at about 30% of the grid points.

## 4  Application to CESM1-LE simulations

In this section, the concept outlined in Sect. 3 is applied to three aspects of precipitation in DJF and JJA (total seasonal precipitation $P$, number of wet days $N_{wet}$, and number of heavy precipitation days $N_{heavy}$), using the CESM1 climate simulations and the WR identification outlined in Sect. 2. To set the scene, Fig. 2 shows the CESM1 seasonal mean precipitation $P$ in the historic period in DJF and JJA, and the corresponding climate change signals $\Delta P$. In DJF, climatological precipitation is largest (within the considered domain) in southwestern Norway (values larger than $10\,\mathrm{mm\,d^{-1}}$) and reaches beyond $7\,\mathrm{mm\,d^{-1}}$ in local maxima off the US east coast and near South Greenland, Iceland, and Scotland (Fig. 2a). The climate change signal in this season (Fig. 2c) shows negative values in the Labrador Sea and the Denmark Strait, with values up to $-2\,\mathrm{mm\,d^{-1}}$, and,





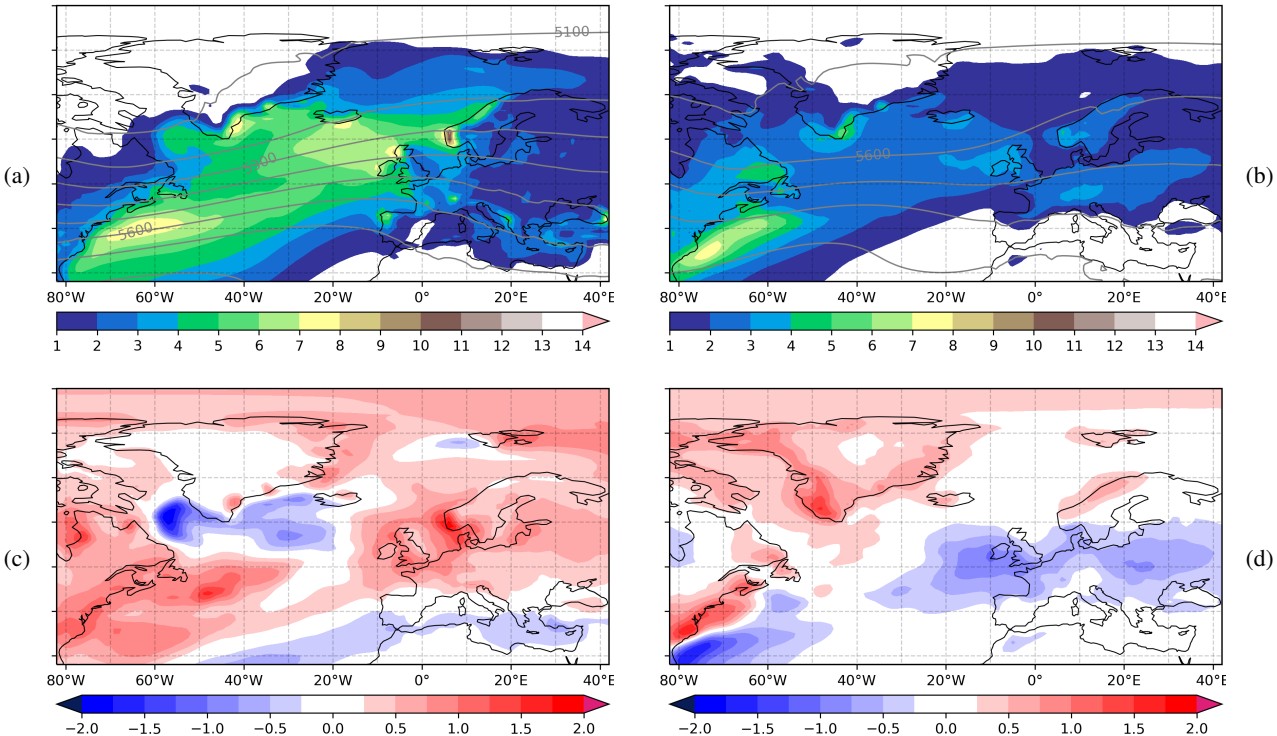

**Figure 2.** Seasonal mean precipitation $P_{hist}$ in the CESM1 historic simulations for (a) DJF and (b) JJA, respectively, and (c,d,) the climate change signal $\Delta P = P_{eoc} - P_{hist}$ in the same seasons. Units are $\mathrm{mm\,d^{-1}}$.

in most of the rest of the domain, weaker positive values with local maxima over the UK and southwestern Norway. In JJA, precipitation is weaker in most parts of the North Atlantic compared to DJF, and the climate change signals are positive along the North American east coast and in Greenland, and negative over most parts of Europe.

To explain the steps required for the calculation of $\gamma_i(P)$, we now first, in Sect. 4.1, assess as an illustrative example the contributions of the frequency and intensity changes of one selected WR to the climate change signal of $P$ in DJF. Then, in Sect. 4.2 and 4.3, results are shown for different WRs in DJF and JJA, and for the three considered aspects of precipitation, respectively.

### 4.1 An illustrative example

As a first example, we consider the frequency and intensity contributions from regime AT (Atlantic Trough) to the climate change signal in $P$ in DJF (Fig. 2c). We have chosen this WR because, in our simulations, it has the largest frequency change in DJF between the two climate periods from $f_{hist,AT}$=13.5% to $f_{eoc,AT}$=15.2% (Fig. 1). For this WR, the relative frequency change of $\frac{\Delta f_{AT}}{f_{hist,AT}} = 12.6\%$ corresponds to the first factor relevant for the calculation of $\gamma_i$ (see Eq. 6). Also, this WR is interesting as it has a strongly positive climate change signal over the British Isles and in the North Sea region (Fig. 3a), i.e., in a region where CMIP6 models show an increase in cyclone track density in DJF (Priestley and Catto, 2022, their Fig. 2e).

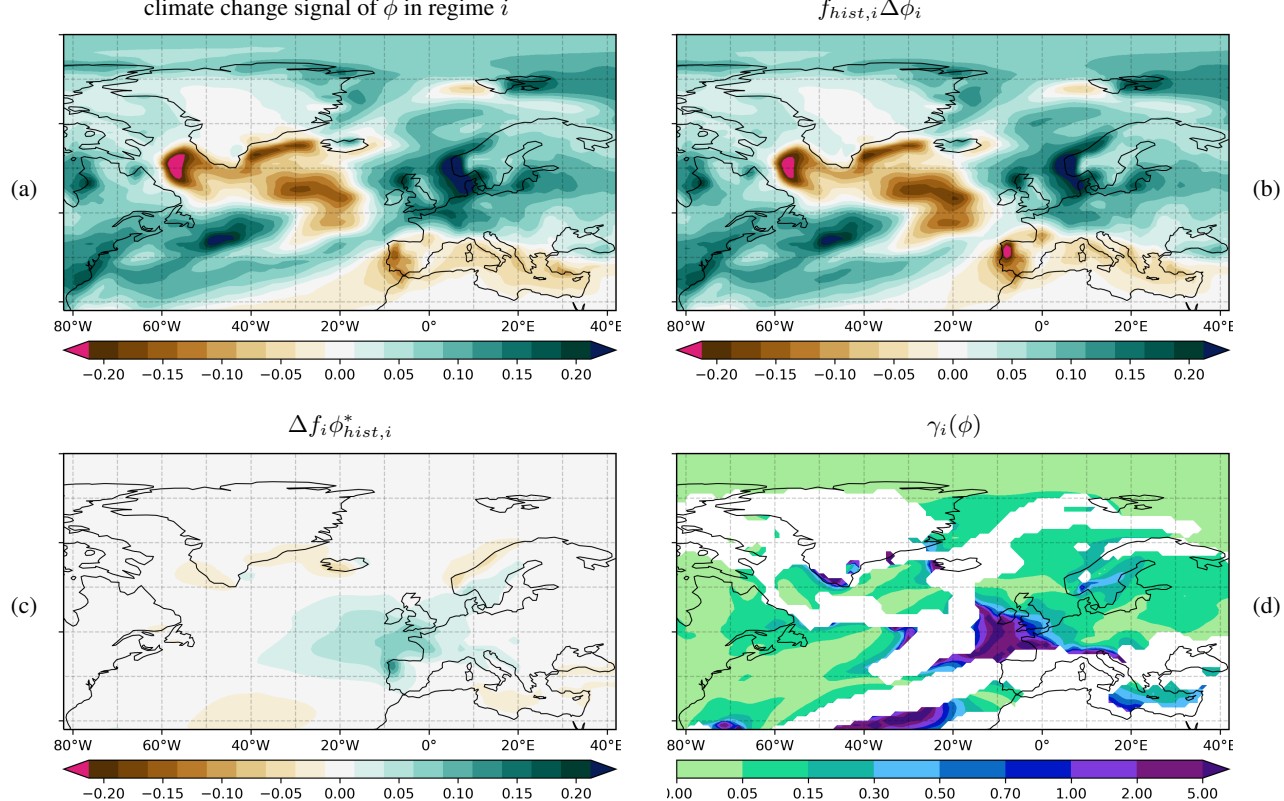

**Figure 3.** **(a)** Contribution of the regime AT to the climate change signal of seasonal mean precipitation $P$ in DJF (in $\mathrm{mm\,d^{-1}}$), i.e., $f_{eoc,AT}P_{eoc,AT} - f_{hist,AT}P_{hist,AT}$; **(b)** contribution to the field shown in **(a)** from intensity changes of $P$ in the regime AT, i.e., term (i) in Eq. (5), $f_{hist,AT}\Delta P_{AT}$ (in $\mathrm{mm\,d^{-1}}$); **(c)** contribution to the field shown in **(a)** from frequency changes of regime AT, i.e., term (iib) in Eq. (5), $\Delta f_{AT}P^{*}_{hist,AT}$ (in $\mathrm{mm\,d^{-1}}$); and **(d)** $\gamma_{AT}(P)$ in dimensionless units.

According to Eq. (5), the total contribution of regime AT to the climate change signal of $P$ in DJF, i.e., $f_{eoc,AT}P_{eoc,AT} - f_{hist,AT}P_{hist,AT}$ (Fig. 3a), can be decomposed into a contribution from intensity changes during regime AT (term (i), Fig. 3b) and frequency changes of regime AT (term (iib), Fig. 3c). Overall, intensity changes clearly determine the pattern shown in Fig. 3a, including the regions with pronounced positive and negative climate change effects on $P$. An exception is the region near the Bay of Biscay, where term (iib) has large positive values exceeding $0.1\,\mathrm{mm\,d^{-1}}$ (Fig. 3c), comparable to those from term (i). The reason for this strong local signal is that the WR-specific precipitation anomaly, $P^{*}_{hist,AT}$, has large positive values in this region, i.e., precipitation is strongly enhanced in this region in regime AT compared to climatology. Peak values of $P^{*}_{hist,AT}$ in the Bay of Biscay exceed $4\,\mathrm{mm\,d^{-1}}$ (Fischer, 2021, their Fig. A.17). Together with the AT frequency change of 1.7% this yields the comparatively large values of term (iib). The high values of $P^{*}_{hist,AT}$ qualitatively agree with the increased frequency of intense precipitation related to atmospheric rivers in this region and WR, as shown by Pasquier et al. (2019).



As a consequence, the key field $\gamma$ investigated in this study, which corresponds to the modulus of the ratio of the fields shown in Fig. 3c and Fig. 3b, attains for this particular example application (total precipitation $P$, season DJF, regime AT) mainly low values of about $\gamma_{AT}(P) \simeq 0.1$, except for the Bay of Biscay region where $\gamma_{AT}(P) > 1$ (Fig. 3d). In some regions the effects of intensity and frequency changes have the opposite sign. For instance, over northwestern Spain, precipitation intensity in

regime AT decreases due to climate change (Fig. 3b) whereas the frequency increase of AT contributes to an increase of $P$ (Fig. 3c) – because of a strongly positive AT-specific precipitation anomaly – resulting in a small negative net decrease of winter precipitation associated with regime AT (Fig. 3a). This first example indicates that most of the climate change signal of $P$ in DJF that can be attributed to regime AT is determined by the field of precipitation intensity change associated with the regime and not due to a frequency change of the regime. However, locally, frequency changes can matter (as shown by values

of $\gamma_{AT}(P)$ on the order of 1 or larger), if they are comparably large (here the relative frequency change amounts to almost 15%) and if the considered regime has a locally pronounced precipitation anomaly $P_{hist,i}^*$.

### 4.2 Systematic analysis of $\gamma_i$ for total precipitation in DJF and JJA

The same procedure, outlined above for the regime AT, can now be applied to all WRs in DJF, in order to obtain the full decomposition of the climate change signal of $P$ into WR-specific intensity and frequency changes. Figure 4 shows the results

for the regimes AR, EuBL, ZO, and, for comparison, again AT. The patterns of the total WR contribution to the climate change signal (left column) show mainly similarities (e.g., negative values in the Labrador Sea and in most of the Mediterranean, and positive values over the eastern USA and the region extending from the UK to the Baltic Sea), but also differences, for instance south of Iceland and over the Iberian Peninsula. As discussed for regime AT in the previous section, these contributions are mainly, or even almost entirely, due to WR-specific intensity changes (2nd column). The role of frequency changes of WRs,

which was discussed to be locally important for regime AT, is (much) smaller for all other regimes (3rd column) and therefore, for regime ZO (as well as for the regimes that are not shown), the values of $\gamma_i(P)$ hardly exceed 0.1 (right column), indicating that intensity changes are at least 10 times more important than frequency changes for explaining climate change signals of $P$ in these regimes. For AR, $\gamma_{AR}(P)$ reaches values above 0.5 southwest of Iceland. The reasons for the lower $\gamma$-values over Europe for the regimes AR, EuBL and ZO compared to AT are twofold: these regimes have (much) lower frequency changes

(Fig. 1) and, except for AR, weaker WR-specific anomalies $P_{hist,i}^*$ (Fischer, 2021, their Fig. A.17).

The same analysis for JJA (Fig. 5) shows first of all generally smaller climate change signals compared to DJF (compare panels in left columns). In regions where there is still a substantial climate change signal, i.e., in regions that are not masked in the plots of $\gamma_i(P)$, the analysis reveals mainly low values of $\gamma_i(P)$ (in most regions smaller than 0.3). The exceptions are the high values of $\gamma_{AR}(P) > 2$ in a band stretching from west of Ireland to Denmark, and of $\gamma_{EuBL}(P) > 1$ in a similar region.

These values result for AR from the exceptionally large relative frequency increase of almost 25% (Fig. 1) and a pronounced dry WR-specific anomaly of $P$ in this region (of about $-1.5\,\mathrm{mm\,d^{-1}}$), yielding a (weak) negative climate change signal (Fig. 5e). For EuBL, the smaller relative frequency change and similar WR-specific precipitation anomalies lead to the smaller but still comparatively large values of $\gamma_i(P)$. Except for these two regimes, WR-specific precipitation anomalies of $P$ over Europe tend to be lower in JJA than in DJF (Fischer, 2021, compare Figs. A.17 and A.19), indicating that $P$ in JJA, compared to DJF,







**Figure 4.** Decomposition of climate change signal of seasonal mean precipitation $P$ in DJF for regimes (from top to bottom) AT, AR, EuBL, and ZO. **(a,e,i,m)** Total contributions of the regimes to the climate change signal (in $\mathrm{mm\,d^{-1}}$); **(b,f,j,n)** contributions from WR-specific intensity changes, i.e., term (i) in Eq. (5); **(c,g,k,o)** contributions from WR-specific frequency changes, i.e., term (iib) in Eq. (5); **(d,h,l,p)** $\gamma_i(P)$ in dimensionless units.

differs less between WRs (but is more variable within a specific WR). This translates directly to low values of the frequency change terms (iib) and therefore to low values of $\gamma_i(P)$ in most regimes.

## 4.3 Systematic analysis of $\gamma_i$ for the frequency of wet days and heavy precipitation days in DJF

We now return to the season DJF, but consider two different aspects of precipitation climatology: the decomposition is now applied to the number of wet days $N_{wet}$ and to the number of heavy precipitation days $N_{heavy}$ (as defined in Sect. 2.1). Figure





| climate change signal | $f_{hist,i}\Delta\phi_i$ | $\Delta f_i \phi^*_{hist,i}$ | $\gamma_i(\phi)$ |
|---|---|---|---|



**Figure 5.** As Fig. 4 but for seasonal mean precipitation $P$ in JJA.

6 shows the results for $N_{wet}$, for the same four WRs as shown previously for $P$. First, we note that the climate change signals of $N_{wet}$ (left column) are qualitatively similar to those for $P$, i.e., seasonal-mean wetting in a particular WR goes typically along with more wet days in the same WR, and vice versa for drying. The exception here is the western North Atlantic where the climate change signal of $N_{wet}$ is close to zero, despite a strong increase of $P$. And as for $P$, also for $N_{wet}$ a large part of the climate change signal is explained by the "intensity changes" (second column in Fig. 6). This means that, for instance

the reduction of the contribution from regime AT to the number of wet days in the Mediterranean (Fig. 6a) is mainly due to a reduced rate of wet day occurrence in regime AT in this region. Consequently, values of $\gamma_{AT}(N_{wet})$ again tend to be small (less than 0.3) in most regions with the exception of the North Sea and Baltic Sea where $\gamma_{AT}(N_{wet})$ reaches values larger than 1, indicating that in this region and WR, WR frequency changes matter more than changes in the rate of wet day occurrence





**Figure 6.** Decomposition of climate change signal for the number of wet days $N_{wet}$ in winter (DJF) for regimes as labeled on the left-hand side of each row. The first three columns show values in wet days per day, otherwise plotting conventions are identical to Fig. 4.

for explaining the climate change signal in the number of wet days. Considering all WRs shown in Fig. 6 and comparing with the same analysis for $P$, the $\gamma_i(N_{wet})$ values appear to be slightly larger.

In stark contrast, for the number of heavy precipitation days shown in Fig. 7, the climate change signals are positive in almost the entire North Atlantic European region, but the gamma values are small for all WRs and hardly exceed values of 0.15. This shows that for the frequency of heavy precipitation days, i.e., for a characteristic of the precipitation climatology that is strongly determined by thermodynamics, WR frequency changes play an even smaller role than for, e.g., the frequency of wet days.







**Figure 7.** Decomposition of climate change signal for the number of heavy precipitation days $N_{heavy}$ in winter (DJF) for regimes as labeled on the left-hand side of each row. The first three columns show values in heavy precipitation days per day, otherwise plotting conventions are identical to Fig. 4.

## 5 Conclusions

In this study, we introduced a framework that helps quantifying the role of changes in the frequency of WRs to climate change signals in surface precipitation. This framework is conceptually simple, and can be applied to any meteorological variable, WR definition, and climate change signal from any climate model and climate change scenario. As a key variable, we suggest to calculate the parameter $\gamma_i(\phi)$, which corresponds, for a specific WR $i$, to the ratio of the contribution from WR frequency changes to the climate change signal of $\phi$ to the contribution of WR intensity changes. A value of 1 indicates that both





contributions are equally important. The derivation of $\gamma_i(\phi)$ shows that, for a given WR $i$, it depends on three independent factors: (i) the relative frequency change of the WR $i$ due to climate change, (ii) the climate change effect on the intensity of the parameter $\phi$ in the WR $i$, and (iii) the anomaly of $\phi$ in WR $i$, which depends on the skill of the WR classification to separate
contrasting situations in terms of the parameter $\phi$.

When applying this framework to seasonal-mean precipitation in the North Atlantic-European region in DJF and JJA in CESM1-LE simulations under the RCP8.5 scenario, and using a 7-WR classification, it turns out that values of $\gamma_i(P)$ are typically small ($< 0.3$). Only for few WRs and in comparatively small regions $\gamma_i(P)$ reached values beyond 1, indicating that in individual WRs the frequency changes can matter in regions where these WRs strongly modulate precipitation occurrence
(i.e., where the WR classification is particularly skillful). However, from the field $\gamma_{overall}(P)$ (Eq. 7, see Fig. 8a), which is less than 0.3 in the entire domain, the main conclusion is that when considering the effects of intensity and frequency changes aggregated across WRs, then indeed, to first order, WR frequency changes are not relevant for explaining climate change signals in $P$. The same fields for the number of wet and heavy precipitation days, $\gamma_{overall}(N_{wet})$ and $\gamma_{overall}(N_{heavy})$, are shown in Fig. 8b,c, and they confirm the results for $P$. Consistent with the discussion in the previous section, $\gamma_{overall}(N_{wet})$
has slightly larger values than $\gamma_{overall}(P)$ in northern Europe, and the values of $\gamma_{overall}(N_{heavy})$ are consistently lower, hardly exceeding 0.05.

There is a combination of reasons why, in our analysis, WR frequency changes only play a minor role. Or, to put it differently, there are three ways how, with other climate simulations and/or WR classifications and/or variables $\phi$ larger $\gamma_i(\phi)$ values could be obtained. These three ways correspond to the three factors discussed in Sect. 3:

– larger relative frequency changes of WRs – in our case, they are typically less than 10% and only for regime AR in JJA reach almost 25%,

– smaller climate change effects on the intensity of $\phi$ – these effects are large for $\phi = P$, $\phi = N_{wet}$, and $\phi = N_{heavy}$, but might be smaller for other variables, and

– larger WR-specific anomalies of $\phi$, i.e., an increased skill of the WR classification in predicting the variable under
consideration.

It is worth further discussing the last of these factors. In the rigorous conceptual approach we take here there is a very explicit dependency of the result whether WR frequency changes are relevant or not on the WR classification. Therefore, a comprehensive answer to the question how important WR frequency changes are for climate change signals should necessarily include a variety of WR classifications. An upper-bound to the importance of WR frequency changes would be reached for
the "best possible" WR classification, i.e., the WR classification with the largest regime-specific anomalies or the highest skill. A measure for assessing the skill of a WR classification in stratifying precipitation is the Brier skill score (Schiemann and Frei, 2010). It quantifies how much more skillful a WR classification is for predicting daily precipitation, compared to assuming climatological precipitation values. Calculating the Brier skill score for the 60th quantile precipitation threshold and the WR classification used in this study yields values of about 0.2 in western Europe and lower values in eastern Europe in





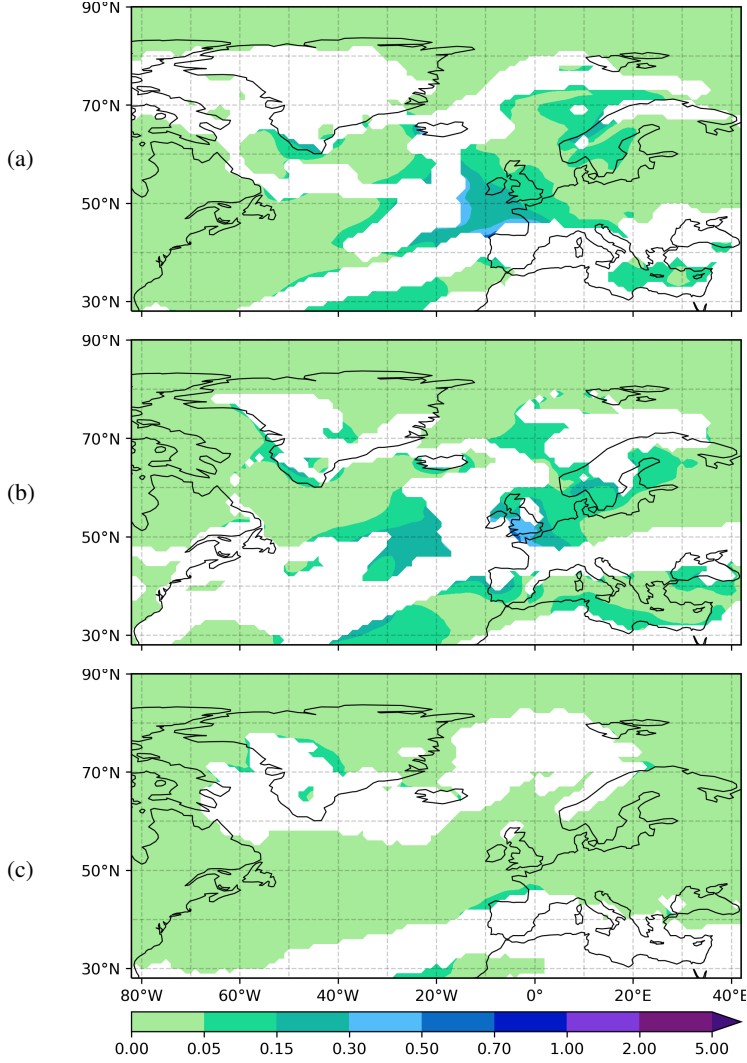

**Figure 8.** The overall $\gamma_{overall}$ in DJF as defined in Eq. 7 for (a) seasonal mean precipitation $P$, (b) the number of wet days $N_{wet}$, and (c) the number of heavy precipitation days $N_{heavy}$. Plotting conventions are identical to the right panels in Fig. 4.

DJF (Fischer, 2021, their Fig. 4.17). Comparison with the Brier skill scores of more than 70 WR classifications for the 60th quantile of daily precipitation in DJF as reported by Schiemann and Frei (2010), which are between 0.1 and 0.3, indicates that the skill of our WR classification is within this range. Therefore, it is unlikely that a different WR classification would yield much larger values of $\gamma_i(P)$ in DJF. In JJA, however, the Brier skill score of our Atlantic basin-wide WR classification for the same precipitation quantile is low with values of less than 0.02 over most of central and eastern Europe.

410       There are few exceptions to the general result of small values of $\gamma_i(P)$. In some regions in western Europe and for WRs with comparatively large frequency changes, $\gamma_i(P)$ reaches values beyond 1. This geographical preference of larger $\gamma_i(P)$ values in





western Europe is related to the fact that precipitation in these regions differs rather strongly between the WRs. In other words, in these regions, the WR classification explains more of the rainfall variability and therefore, circulation changes as quantified by changing WR frequencies, can matter for explaining WR-specific climate change signals in $P$. As shown in Fig. 8, near the

British Isles $\gamma_{overall}(P)$ reaches its largest values of about 0.3, indicating that in this part of the North Atlantic storm track in DJF, WR frequency changes also contribute about one fourth to the overall increase in $P$ in the considered climate change simulations (Fig. 2c).

However, the resulting values of $\gamma_i(\phi)$ not only depend on the regime classification, but clearly also on the climate model and the climate change scenario, on the region, and on the parameter $\phi$. We hope that the conceptual analysis and examples

shown in this study will motivate others to assess $\gamma_i(\phi)$ for different WR classifications, in other regions, and with output from other climate simulations. In terms of different variables, we briefly compared results for three aspects of precipitation and found slightly larger values of $\gamma_i(\phi)$ for the number of wet days $N_{wet}$ than for seasonal-mean precipitation $P$, and the lowest values of $\gamma_i(\phi)$ for the number of heavy precipitation events $N_{heavy}$. These differences might be related to the fact that some meteorological variables are affected strongly by climate change through thermodynamic effects, e.g., the intensity of heavy

precipitation events (Pendergrass, 2018) or the frequency of days above a certain absolute temperature threshold. The larger the thermodynamic effect of global warming on a certain variable, the larger (and the more uniform across weather regimes) is the intensity change $\Delta\phi_i$ – and thus the smaller is $\gamma_i$. Conversely, for variables for which thermodynamic arguments constrain less their climate change signal, no physical reason exists for why $\Delta\phi_i$ should be large (and uniform across weather regimes), and thus we expect larger values of $\gamma_i(\phi)$ for such variables.

Several studies explicitly quantified dynamic vs. thermodynamic contributions to climate change. For instance Pfahl et al. (2017) found that the dynamic contribution to changes in extreme precipitation is mainly important in the subtropics but not in the midlatitudes. This result is qualitatively consistent with our finding that $\gamma_i(N_{heavy})$ is small in the North Atlantic-European domain. However, it is important to mention that our decomposition into WR frequency and intensity changes cannot be easily compared to this alternative approach of separating thermodynamic and dynamic effects. WR frequency changes can

be essentially regarded as dynamic contributions, whereas WR intensity changes most likely have a strong thermodynamic contribution, but may also contain a substantial effect from dynamics. For instance, in a warmer climate, intense cyclones may become deeper due to enhanced precipitation and diabatic heating (e.g., Büeler and Pfahl, 2019; Sinclair et al., 2020; Dolores-Tesillos et al., 2022; Binder et al., 2023), and this dynamic contribution would be quantified in our approach as a WR intensity increase (assuming here that the intense cyclones still occur in the same WR). It is therefore important to improve in

future studies the understanding of the WR intensity changes. While $\Delta\phi_i(P)$ is fairly uniform across WRs (see 2nd column in Fig. 4), which most likely indicates a strong thermodynamic control on these fields, the WR-specific intensity changes for the frequency of heavy precipitation events, $\Delta\phi_i(N_{heavy})$ (2nd column in Fig. 7), is much less uniform, which is likely related to dynamical effects such as the preferred location of intense cyclones in different WRs.

In future studies, the conceptual $\gamma$ approach could be extended to decompositions based on the frequency of occurrence

and intensity of synoptic weather systems such as cyclones, anticyclones and fronts. That is, instead of asking how WR frequency changes affect climate change signals of, e.g., precipitation, one could diagnose the relevance of frequency changes



in the occurrence of cyclones and fronts. Examples of such weather system decomposition were presented in Yettella and Kay (2017) and in Chang et al. (2022). The approach introduced in this study could provide useful guidance for the analysis and interpretation of how weather system frequencies and intensities affect trends in surface weather parameters.

*Code and data availability.* For ERA-Interim reanalyses (Dee et al., 2011), the Public Datasets Service closed on 1 June 2023. However, access to the ERA-Interim and CESM-LE fields used in this study are available from the authors upon request. The code of the CESM model (Hurrell et al., 2013) used for the CESM-LE simulations is available from https://www.cesm.ucar.edu/models/cesm1.0/ (last access: 2 April 2024).

*Author contributions.* LF, MR and HW designed this study, and LF performed all analyses with support from DoB, CMG and MR. LF, DoB
and HW wrote the manuscript, with feedback about the results and text from all co-authors.

*Competing interests.* At least one of the (co-)authors is a member of the editorial board of Weather and Climate Dynamics.

*Acknowledgements.* DoB acknowledges funding from the Schweizerischer Nationalfonds zur Förderung der Wissenschaftlichen Forschung (project 205419) and MR from the H2020 European Research Council (INTEXseas, grant no. 787652). The contribution of CMG was funded by the Helmholtz Association as part of the Young Investigator Group "Sub-seasonal Predictability: Understanding the Role of
Diabatic Outflow" (SPREADOUT, grant VH-NG-1243)."



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
