# Peer review of "How relevant are frequency changes of weather regimes for understanding climate change signals in surface precipitation in the North Atlantic-European sector? – a conceptual analysis with CESM1 large ensemble simulations"

_EGUsphere, 2024_

## Referee Comment (RC2)

Review of "How relevant are frequency changes of weather regimes for understanding climate change signals in surface precipitation in the North Atlantic-European sector? – a conceptual analysis with CESM1 large ensemble simulations" by Luise Fischer et al.

The paper focuses on the decomposition of the precipitation response with climate change in terms of different weather regimes and defines a novel metric to quantify the relative importance of dynamic (related to regime frequency) vs thermodynamic (related to regime intensity) changes. Although the metric is not able to definitely disentangle the two components, the changes in regime intensity dominate for most of the domain, indicating that regime frequency changes are of secondary importance to understand future changes in surface precipitation.
I think the results are very interesting and relevant for the community. The paper is well written, with attention to the details, denoting a great accuracy both in the work and in the presentation of the results. I invite the authors to consider a couple of doubts regarding the methodology and a minor issue regarding the assessment of the significance of the results, along with some more specific comments.

**General comments**

- Significance of the results. I would appreciate a brief discussion of the significance of the changes in the composites with respect to the internal variability. For example, considering the standard deviation of 10-year chunks in the historical period would give an estimate of the variability of the regime-specific composites. This could also be a more quantitative metric to assess the skill of the regime in decomposing the precipitation field (e.g. comment at line 260). Also, this relates to comment at line 277.

- Removal of the climatology. The authors remove the climatology of the two periods separately to get the geopotential anomalies, which are then projected on the observed ERA5 regimes.
Since the anomalies refer to different mean states, this may have an impact on the actual dynamical configuration of the regimes and hence on the precipitation composites. Out of curiosity, have you checked whether the composites of some more dynamical field differ between corresponding eoc and hist regimes (e.g. the zonal wind)? In that case, this may constitute a dynamical effect which is now by construction inside the "regime intensity" component (related to discussion at lines 435-443).

- The intensity change term (i) in equation 5 contains not only the change in the regime-specific precipitation anomaly ($Phi^*_{eoc,i}$ - $Phi^*_{hist,i}$) but also the change in the overall precipitation climatology between the two periods ($\underline{Phi}_{hist}$ - $\underline{Phi}_{eoc}$). The climatology part has instead been removed from term (ii) since it would have summed to zero over all regimes (term iia), so I was wondering whether this creates by construction an asymmetry between the two terms that make up gamma (making gamma smaller). It may be helpful to see the contribution of the climatology separated from the regime-specific anomalies in term (i) (so e.g. subtracting the field in Fig. 2c from the second column in Fig. 4).

**Specific comments**

L45. The matter is quite debated, I would be softer on affirming that there is "clear" evidence that the regimes "represent physical modes of the atmosphere". For example, the number of regimes to be considered varies in literature, and there is no clear indication that 7 is the "true" number (to my knowledge). I agree that the atmospheric flow shows a tendency for non-linearity and preferred

states, but I also think that the classification in regimes is always artificial to a certain extent. I would suggest the review by Hannachi et al. (2017) for an historical perspective on this point.

Hannachi, Abdel., David M. Straus, Christian L. E. Franzke, Susanna Corti, and Tim Woollings. "Low-Frequency Nonlinearity and Regime Behavior in the Northern Hemisphere Extratropical Atmosphere." *Reviews of Geophysics* 55, no. 1 (2017): 199–234. https://doi.org/10.1002/2015RG000509.

L51. I would suggest to add Madonna et al. (2021), which discuss the link between various regime frameworks and seasonal precipitation/temperature anomalies in Europe.

Madonna, Erica, David S. Battisti, Camille Li, and Rachel H. White. "Reconstructing Winter Climate Anomalies in the Euro-Atlantic Sector Using Circulation Patterns." *Weather and Climate Dynamics* 2, no. 3 (August 25, 2021): 777–94. https://doi.org/10.5194/wcd-2-777-2021.

L66. The result by Huguenin et al. (2020) should be moved inside the discussion at lines 74-78 regarding the "rather low consensus on future changes in WR occurrence". Also, regarding this point, it may be worth discussing other evidence of circulation changes (in winter) in warmer climates, for example the work by Oudar et al. (2020) and Peings et al. (2018).

Oudar, Thomas, Julien Cattiaux, and Hervé Douville. "Drivers of the Northern Extratropical Eddy-Driven Jet Change in CMIP5 and CMIP6 Models." *Geophysical Research Letters* 47, no. 8 (2020): e2019GL086695.

Peings, Yannick, Julien Cattiaux, Stephen J Vavrus, and Gudrun Magnusdottir. "Projected Squeezing of the Wintertime North-Atlantic Jet." *Environmental Research Letters* 13, no. 7 (2018): 074016.

L156. How is the seasonal normalization coefficient computed for the historical and future model fields? I expect the coefficient would be different from the one computed for the ERA reanalysis, and would differ for the future and historical periods too. How does this choice impact the seasonality of the regimes and their projection on the ERA EOF space?

L165. What is the sensitivity to this threshold? i.e. what was the proportion of no-regime days with the original threshold of 1?

L176-180, Figure 1. Even if this is a single model study, a quick comment and comparison of these results with others in literature would be needed here.

L233. The sign of the ratio may be also interesting to investigate (i.e. do dynamic and thermodynamic effects contribute in the same direction?).

L242. The order of magnitude estimate is fine, but I would avoid a more quantitative estimate at this stage (e.g. 10 -> 9-11%), since the frequency change could in principle be much larger.

L260. For a "very good WR classification".. I understand that you expect delta Phi for each regime to be small with respect to the typical regime anomaly of Phi. However, the estimate is quite arbitrary (in principle, the regime intensity could increase/decrease by a more significant fraction) and a low value of this ratio could be due to other causes, rather than to the quality of the regime classification.

L277. I think it would be more appropriate here to set a more objective threshold on the climate change signal, e.g. where the signal is significant with respect to internal variability (e.g. considering the historical variability in a 10-year random sample). What does the 30% threshold correspond to in terms of climate change signal?

Fig. 2. I suggest to use the green/brown colorbar for the precipitation differences, as done in other figures. Also, it would be nice to add a title with the respective seasons on top (just a suggestion).

L286. Also, negative values are found in the Mediterranean region, it may be worth commenting on that.

L335. "weaker WR-specific anomalies". I think it would be worth adding in a supplementary a figure with the P* fields (hist and eoc, and/or the delta P fields), to allow comparison of the relative magnitude before multiplying by additional factors. Fig. A.17 and A.19 of the doctoral thesis would be very useful here.

L345. "but is more variable within a specific WR". I don't think this is indicated by the precipitation anomaly composite.

L435. I appreciate the observation regarding the fact that intensity changes could also contain a dynamical signal, for example in terms of the amplitude of the dynamical anomalies. Do you see a way to extract some more information out of this, i.e. to separate a general thermodynamic response to regime-specific features?

L440. "delta Phi is fairly uniform across WRs". This is not true for all regions, for example the central North-Atlantic and the iberian peninsula.

---

## Author Response (AR1)

*Paper egusphere-2024-1253*

**How relevant are frequency changes of weather regimes for understanding climate change signals in surface precipitation in the North Atlantic-European sector? – a conceptual analysis with CESM1 large ensemble simulations**

by Luise J. Fischer, David N. Bresch, Dominik Büeler, Christian M. Grams, Matthias Röthlisberger, and Heini Wernli

*Replies*

We are most grateful to the reviewers for their thoughtful and constructive comments that helped us to further improve the manuscript. Based on the reviewers' suggestions, we implemented several changes in the manuscript. The main changes are that:

- We better explained certain aspects of our rationale, which in the previous version led to some confusion with the reviewers.
- We added additional panels/information to the figures to better illustrate some of the important basics of our study: the regime-specific precipitation anomalies in Fig. 3c and Fig. 4-7 (third column), and the wet day climatology in Fig. 2.

Below we provide a one-to-one response to all points raised by the reviewers. The reviewers' comments are in black and our replies in blue. Line numbers refer to the revised version of our paper.

**Reviewer 1**

Recommendation: minor revisions

This paper provides a method to assess the relative contributions of the weather regimes frequency and intensity changes as well as of the skill of the weather regime classification to the precipitation change by the end of the century using reruns of the CESM1 Large Ensemble. The authors find that the change in frequency has a minor role except for some regimes in very specific regions. The manuscript is well written and relatively clear. My most major comment concerns Section 3 and the description of $\gamma$. All my comments can be found below in the order of the manuscript followed by some technicalities.

Introduction:

Lines 90-99: point 2 sounds like a repetition of point 1 because "quantify the relevance of WR frequency changes" is already mentioned in point 1. Therefore, point 2 could just be 'use $\gamma_i(\Phi)$ on precipitation $\Phi=P$, $\Phi=N_{wet}$ and $\Phi=N_{heavy}$'. The details about the model used and the choice of seven regimes can come later in section 2.

We agree, thank you, we modified the formulation of the 2$^{nd}$ main objective (L95). The 2$^{nd}$ objective now reads "2. quantify $\gamma_i(\Phi)$ for mean precipitation $\Phi=P$, the number of wet days $\Phi=N_{wet}$, and the number of heavy precipitation days $\Phi=N_{heavy}$ based on large ensemble climate simulations and a specific WR classification."

Section 2.3:

Lines 144-151: the authors point out that performing the weather regime classification on the CESM simulations historical and end-of-century periods separately would lead to different weather regimes. What about the ability of CESM in representing the historical weather regimes? The authors do not mention this aspect, but it could also be a reason for using the weather regimes patterns from ERA-Interim. Have the weather regimes in CESM been studied before? I would appreciate if the authors could add here a sentence on this topic.

When we wrote "… would lead to different weather regimes" we did not mean that they need to be substantially different, but rather that they are not identical. We agree that there are several ways, how a weather regime analysis could be done with climate simulations. We decided to stick to the regimes as identified in the reanalyses – which we regard as a good reference given the observational basis of reanalysis data – and project the simulated flows to these regimes. An alternative approach would be to perform an EOF analysis with fields from the climate simulations and then determine regimes separately in the present-day and future climate. This would be quite some work and, as we briefly explain in the manuscript, it would make quantifying the climate change effects on regime frequency more cumbersome because there would be a change in the regime frequencies (what we are interested in) but most likely also in the details of the regime patterns. We therefore did not identify weather regimes based on a separate EOF analysis in CESM. However, Fischer (2021) showed that, when using our

method of identifying weather regimes in CESM simulations, the mean seasonal and year-round regime frequencies in CESM historical simulations are similar to the corresponding frequencies in ERA-Interim, indicating that the large-scale flow variability over the North Atlantic European region is well captured in the CESM simulations. Furthermore, we recommend looking at the study by Fabiano et al. (2021; https://doi.org/10.5194/wcd-2-163-2021), to get an impression about how well state-of-the-art climate models reproduce Atlantic-European weather regime patterns.

Section 3:

Lines 228-264: I do not understand why the authors spend so much time describing the two ratios $\Delta f_i/f_{hist,i}$ and $\Delta\Phi_i/\Phi^*_{hist,i}$, as written in Eq. (6), when the rest of the paper and the figures deal with $\Delta f_i\ \Phi^*_{hist,i}$ and $\Delta\Phi_i\ f_{hist,i}$. Also, it is confusing when comparing with $\gamma_{overall}$, which actually uses the terms (iib) and (i) from Eq. (5). I found this part difficult to follow and confusing in light of the rest of the paper.

It is unfortunate that this part was confusing to the reviewer, as it is, in our view, a central part of the paper. We guess that the insertion of Eq. (7) about $\gamma_{overall}$ might have contributed to the confusion, as it interrupts a bit the flow between Eq. (6) and the longer discussion of the three factors that contribute to $\gamma_i$ in L237-264. We therefore moved Eq. (7) to after this discussion. The reason why we discuss the three factors in some detail is that they are, as we think, in a nice way independent: the first factor, $\Delta f_i/f_{hist,i}$, is about the frequency change of the regime. It depends on the regime classification and the intensity of climate change but is a constant factor that does not vary in space. The other two factors are spatially variable, i.e., they determine the geographical pattern of $\gamma_i$. The 2$^{nd}$ factor depends on the intensity of climate change, and the 3$^{rd}$ factor on the "skill" of the regime classification to distinguish (in our case) between wet and dry conditions. We regard it as insightful and important to discuss $\gamma_i$ in this way, and the expression in Eq. (6) elegantly separates the frequency and intensity aspects. What we then show in the figures is mathematically identical, and since $\gamma_i$ as written in Eq. (6) is in the end simply the ratio of terms (i) and (iib) in Eq. (5), it is, in our view, fully consistent to show these two terms in the figures. We mention this again in the revised version in L315 and are confident that in this way we can avoid confusion.

Line 253: "If days were randomly attributed to one of the WRs": This sentence confused me at first because I did not which "days" the sentence referred to and "one of the WRs" made me think that only one weather regime was used here. I suggest rewriting the other way around like: 'If each weather regime was attributed to a different random set of days within the historical period'.

Thank you for pointing out that this sentence was not clear. We think that the main problem is the formulation "to one of the WRs", and we rewrote the sentence in L247 as "If each day within the historical period was randomly attributed to a WR, …".

Lines 277-278: Could the authors add the actual values of these thresholds, and which variable is used to define those thresholds? I suspect that the authors use Fig. 2c,d to

determine the threshold for Figs. 3, 4, and 5. Moreover, the percentage given here (30%) is quite vague as the reader does not know if the authors mean 30% of the grid points within the domain plotted (~30°N-90°N / ~80°W-~40°E) or within the Northern Hemisphere (or even globally).

We meant 30% of grid points within the domain plotted, and we clarified this (L273). The reviewer is correct that we use Fig. 2c,d to determine the thresholds for all subsequent figures. For instance, the grid points masked in Fig. 3d are the about 30% grid points shown in Fig. 2c with the smallest absolute climate change signal in precipitation (mainly the white regions in Fig. 2c). We prefer not to add the threshold values to the paper, as this would put too much weight on these pragmatically chosen values. The only aim of this masking is to avoid that the reader starts interpreting large values of $\gamma_i$ in regions where the question whether frequency changes of regimes contribute to the climate change signal, is not relevant, simply because the signal is small.

Section 4:

Line 286: "Denmark Strait" I would rather locate the negative response over the Irminger Sea. Please check if you agree and eventually correct.

Thanks, much better, we changed "Denmark Strait" to "Irminger Sea" (L285).

Figure 2: It would be great to also have the DJF climatology of wet days and heavy rain days as well as their response to climate change. Four more panels could be added to this figure. Moreover, how do these DJF and JJA precipitation climatologies compare to reanalyses or observation-based products? Could the authors add a sentence on this?

Thanks for this suggestion. We included additional four panels to Fig. 2 as suggested. About the precipitation climatologies: The DJF and JJA precipitation climatologies in CESM compare reasonably well with ERA5 and with the climatology of Xie and Archin (1997, their Figs. 11 and 13). For instance, in DJF, Xie and Arkin showed values of about 5 mm day$^{-1}$ south of Iceland, 3.5 mm day$^{-1}$ over Ireland, and 2.5 mm day$^{-1}$ over the Alps, which is a bit lower than the values shown in Fig. 2a in these regions (about 6.5, 5, and 4 mm day$^{-1}$, respectively). The higher resolution ERA5 reference dataset has values of about 6, 4, and 3.5 mm day$^{-1}$ in the three selected regions (not shown). This indicates that the precipitation climatology in CESM agrees reasonably well with reference climatologies, and, in particular, captures large-scale gradients from the North Atlantic to central Europe.

Line 287: "weaker positive values" I do not find them that weak. I suggest to add "slightly" before "weaker".

Thank you, changed as suggested (L286).

Figure 3a: Why does this figure look so much like the DJF response displayed in Fig. 2c? Also, it seems that all four weather regimes presented here change in almost the same way in

the future. Why is that? Somehow, I would have expected more WR-specific changes, meaning following the WR precipitation anomalies shown in Fischer (2021). Could the authors comment on that aspect? Is it expected?

No, this was not expected! Indeed, the climate change signal is, at least at first sight, very similar in all regimes, and we were also surprised by this. It means that the climate change signal is, at least in some regions, mainly determined by other factors than the circulation variability captured by the weather regimes. A prominent example is the strong decrease of precipitation in the Labrador Sea in all regimes. This is caused by the poleward retreat of the sea ice edge in the future climate, which will strongly reduce shallow convection associated with intense cold air outbreaks south of the sea ice edge in the current climate. However, closer inspection of the left column panels in Fig. 4 also shows important differences between the regimes: (i) south of Iceland, the climate change signal is either strongly negative, strongly positive, or close to zero in the regimes AT, AR, and EuBL, and (ii) also in Spain and along the west coast of Scandinavia there are large differences between the regimes. We now discuss these first-order similarities and detailed differences briefly in the revised version (L340).

Line 308: I find slightly annoying to have to look for Fischer (2021) to find the precipitation anomalies associated with the weather regimes. Would there be a way to include this information as contours on panel (a) of Fig. 3 or on panel (c) since this panel is less busy than the others, for example? Or could these figures be added to a supplement?

Apologies for not including these fields explicitly in the first submission, and thanks for your interest to look at them! These fields were to a certain degree visible in Fig. 3c and in the third column of Fig. 4, but because of using the same contour intervals as in Fig. 3b (for good reasons), not much is visible of the regime-specific precipitation anomalies. We therefore decided to add a few contours of $\Phi^*_{hist,i}$ to these figures (see new Fig. 3c and third columns in Fig. 4-7).

Line 335: "weaker WR-specific anomalies". To me it looks like the anomalies in Fig. A17 in Fischer (2021) are also strong for the other weather regimes. The zonal weather regime exhibits a strongly positive anomaly and the European Blocking a strong negative anomaly. Therefore, I suggest to modify this sentence.

Thanks, we mainly agree and simplified the sentence. It now reads (L346) "The reasons for the lower γ-values over Europe for the regimes AR, EuBL and ZO compared to AT are mainly the (much) lower frequency changes in these regimes (Fig. 1)."

Line 341: "of about 1.5 mm day$^{-1}$" This value can be found in Fischer (2021), right? If yes, please add the reference. If not, please write where I can find these values.

Yes, the value was from Fig. A17 in Fischer (2021). Since we now show contours of $\Phi^*_{hist,i}$ in some panels, we refer to them (L356).

Technical comments:

Line 238: "how large are the fields γi(Φ)" → "how large are the fields γi(Φ)?"

Changed as suggested (L232).

Figure 2: the latitude labels are missing on all panels. The gray contours and their labels in panels (a) and (b) are quite difficult to see. Please consider using another color and to not overlay the contour on its label. Moreover, those lines are not described in the caption. I suppose they are the 500-hPa geopotential height (in m).

We added latitude labels to the panels, and we now mention the 500-hPa geopotential height contours in the caption. Since this field is of secondary importance, we would like to keep the gray contours but we avoid that the labels overlay the contours.

Figure 3: I suggest to replace the "i" by AT so that we can immediately see that this figure is about the Atlantic trough. Moreover, in panel (d), the colorbar label 0.00 and longitude label 40°E are cut at the edges of the figure.

In the headings of the four panels, we everywhere replaced "i" by "AT" and mention in the caption that Φ=P. Also, the labels at the edges of the panels are no longer cut.

Figures 4, 5, 6, and 7: there is a slight misalignment between the left column and the other columns as visible from the colorbars that are higher in the left-most column compared to the other columns.

Thanks for spotting this. We corrected this.

Line 388: add a comma between "Φ" and "larger".

Changed as suggested (L400).

Lines 426-427: the larger [...] is the intensity change $\Delta\Phi_i$ – and thus the smaller is $\gamma_i$ → the larger [...] the intensity change $\Delta\Phi_i$, the smaller $\gamma_i$

Changed as suggested (L439).

References: many references have a double slash (//) after doi.org. The doi is missing for references in lines 483, 519, 553, and 576.

Thanks for spotting this, all doi were corrected.

**Reviewer 2**

Recommendation: minor revisions

The paper focuses on the decomposition of the precipitation response with climate change in terms of different weather regimes and defines a novel metric to quantify the relative importance of dynamic (related to regime frequency) vs. thermodynamic (related to regime intensity) changes. Although the metric is not able to definitely disentangle the two components, the changes in regime intensity dominate for most of the domain, indicating that regime frequency changes are of secondary importance to understand future changes in surface precipitation.

I think the results are very interesting and relevant for the community. The paper is well written, with attention to the details, denoting a great accuracy both in the work and in the presentation of the results. I invite the authors to consider a couple of doubts regarding the methodology and a minor issue regarding the assessment of the significance of the results, along with some more specific comments.

General comments

Significance of the results. I would appreciate a brief discussion of the significance of the changes in the composites with respect to the internal variability. For example, considering the standard deviation of 10-year chunks in the historical period would give an estimate of the variability of the regime-specific composites. This could also be a more quantitative metric to assess the skill of the regime in decomposing the precipitation field (e.g. comment at line 260). Also, this relates to comment at line 277.

We are not completely sure whether we understand this suggestion, and we therefore did not do additional calculations. We hope the following comments at least partially contribute to addressing this comment:
- In her PhD thesis (Fischer, 2021; Sect. 4.3.1), the first author did careful statistical testing of the statistical significance of the frequency changes of the regimes and found that the largest frequency changes were statistically significant at the 99% level.
- The study by Yettella and Kay (2017) investigated precipitation changes as simulated by the same CESM1 large ensemble. In their Fig. 1e,f they showed precipitation differences between the periods 2081-2100 and 1986-2005 and their fields look very similar to our difference fields in DJF and JJA (our Fig. 2c,d), although we considered shorter periods (2091-2100 vs. 1990-1999). According to the statistical test performed by Yettella and Kay, the identified differences are statistically significant at the 95% confidence level across the entire North Atlantic European domain, and we therefore refrained from repeating a similar significance test.
- On a more general level, we tend to question the usefulness of performing this sort of significance tests with large ensembles to investigate the null hypotheses that mean precipitation in both periods is identical. Whereas testing significance is meaningful with observations / reanalysis data, where the data volume is naturally constrained, performing

- such tests with ensemble data appears less meaningful because there is no limit in generating more ensemble members. In the limit of very large ensembles, even tiny differences would appear as significant since the power of the statistical test to identify departures from the null hypothesis increases with the sample size.
- Maybe the reviewer was however more interested in quantifying the time of emergence, which addresses the question when changes in mean precipitation emerge from internal variability. For instance, Maraun (2013) quantified emergence for seasonal mean precipitation and found times of emergence prior to 2100 in almost all regions across Europe, both in DJF and JJA. Quantifying these times in our ensemble would be beyond the scope and not add substantially to the objectives of our study.

Maraun, D.: When will trends in Europe mean and heavy daily precipitation emerge? Env. Res. Lett., 8, 014004, https://doi.org/10.1088/1748-9326/8/1/014004, 2013.

Yettella, V. and Kay, J. E.: How will precipitation change in extratropical cyclones as the planet warms? Insights from a large initial condition climate model ensemble, Clim. Dynam., 49, 1765–1781, https://doi.org/10.1007/s00382-016-3410-2, 2017.

Removal of the climatology. The authors remove the climatology of the two periods separately to get the geopotential anomalies, which are then projected on the observed ERA5 regimes. Since the anomalies refer to different mean states, this may have an impact on the actual dynamical configuration of the regimes and hence on the precipitation composites. Out of curiosity, have you checked whether the composites of some more dynamical field differ between corresponding eoc and hist regimes (e.g. the zonal wind)? In that case, this may constitute a dynamical effect which is now by construction inside the "regime intensity" component (related to discussion at lines 435-443).

Thank you for this very interesting comment. First, we would like to mention that with our approach, the WR-specific geopotential height anomalies at 500 hPa are very similar in ERA-Interim and the two climate periods considered with CESM (Figs. A.7-A.10 in Fischer, 2021). We regard this as an important a posteriori check that our method of attributing the WRs in different climates is meaningful. Addressing more specifically the reviewer's question, no, we did not investigate the change in, e.g., zonal wind at 500 or 300 hPa, and we think that such an analysis of the mean circulation changes in the CESM ensemble simulations would require a separate study. In our final author comments, we dared to disagree with the last sentence of the reviewer and wrote that in our view, a mean circulation change does not necessarily affect "regime intensity", but it could still affect "regime frequency". However, we thought about this again and now agree that it is very likely that, as suggested by the reviewer, the mean circulation change influences regime intensity (but still, it could also affect regime frequency). This influence of the mean circulation change on regime intensity is likely a reason why the intensity changes are relatively similar across the regimes. Therefore, this interpretation is consistent with our conclusion that WR frequency changes, i.e., the frequency in patterns of deviations from the mean flow, are of secondary importance for explaining climate change signals in precipitation. The next comment is a direct continuation of this

theme, and our reply explains why we regard our approach as best suited to address the main objective of this study.

The intensity change term (i) in equation 5 contains not only the change in the regime-specific precipitation anomaly ($Phi^*_{eoc,i} - Phi^*_{hist,i}$) but also the change in the overall precipitation climatology between the two periods ($Phi_{hist} - Phi_{eoc}$). The climatology part has instead been removed from term (ii) since it would have summed to zero over all regimes (term iia), so I was wondering whether this creates by construction an asymmetry between the two terms that make up gamma (making gamma smaller). It may be helpful to see the contribution of the climatology separated from the regime-specific anomalies in term (i) (so e.g. subtracting the field in Fig. 2c from the second column in Fig. 4).

Thank you for this comment. We agree with the reviewer that our term (i) contains the overall climate change signal, whereas term (iib) does not. The reason for this asymmetry mentioned by the reviewer is the fact that the sum (over all regimes) of the frequency changes must be zero, whereas the sum of the intensity changes corresponds to the total climate change signal and is not zero. In our view, our approach is best suited to address the specific question of this study, which is "how relevant are frequency changes of weather regimes – i.e. of patterns of flow deviations from the mean – for understanding climate change signals in surface precipitation". If we followed the suggestion by the reviewer (and also split term (i) into a term with the overall precipitation change and a term with a regime-specific precipitation change, $\Delta\Phi_i = \Delta\Phi + \Delta\Phi^*_i$, then we would address a different question namely "how relevant are frequency changes of weather regimes due to climate change relative to changes of regime-specific surface precipitation anomalies", which is not what we are aiming for in this study.

Specific comments

L45. The matter is quite debated, I would be softer on affirming that there is "clear" evidence that the regimes "represent physical modes of the atmosphere". For example, the number of regimes to be considered varies in literature, and there is no clear indication that 7 is the "true" number (to my knowledge). I agree that the atmospheric flow shows a tendency for non-linearity and preferred states, but I also think that the classification in regimes is always artificial to a certain extent. I would suggest the review by Hannachi et al. (2017) for a historical perspective on this point.

Hannachi, Abdel., David M. Straus, Christian L. E. Franzke, Susanna Corti, and Tim Woollings. "Low-Frequency Nonlinearity and Regime Behavior in the Northern Hemisphere Extratropical Atmosphere." Reviews of Geophysics 55, no. 1 (2017): 199–234. https://doi.org/10.1002/2015RG000509.

Thank you, we chose a softer formulation and included a reference to this interesting review article (L47).

L51. I would suggest to add Madonna et al. (2021), which discuss the link between various regime frameworks and seasonal precipitation/temperature anomalies in Europe.

Madonna, Erica, David S. Battisti, Camille Li, and Rachel H. White. "Reconstructing Winter Climate Anomalies in the Euro-Atlantic Sector Using Circulation Patterns." Weather and Climate Dynamics 2, no. 3 (August 25, 2021): 777–94. https://doi.org/10.5194/wcd-2-777-2021.

Thank you, reference has been included (L54).

L66. The result by Huguenin et al. (2020) should be moved inside the discussion at lines 74-78 regarding the "rather low consensus on future changes in WR occurrence". Also, regarding this point, it may be worth discussing other evidence of circulation changes (in winter) in warmer climates, for example the work by Oudar et al. (2020) and Peings et al. (2018).

Oudar, Thomas, Julien Cattiaux, and Hervé Douville. "Drivers of the Northern Extratropical Eddy-Driven Jet Change in CMIP5 and CMIP6 Models." Geophysical Research Letters 47, no. 8 (2020): e2019GL086695.

Peings, Yannick, Julien Cattiaux, Stephen J Vavrus, and Gudrun Magnusdottir. "Projected Squeezing of the Wintertime North-Atlantic Jet." Environmental Research Letters 13, no. 7 (2018): 074016.

Thanks also for making us aware of these studies. However, we did not find a clear link to our discussion of weather regime / circulation type changes, and we therefore did not reference them. However, we moved the sentence about the results by Huguenin et al. (2020) as suggested (L77).

L156. How is the seasonal normalization coefficient computed for the historical and future model fields? I expect the coefficient would be different from the one computed for the ERA reanalysis and would differ for the future and historical periods too. How does this choice impact the seasonality of the regimes and their projection on the ERA EOF space?

We apologise because the formulation about the normalization coefficient was incorrect. Thank you for picking up this detail. The normalization coefficient is computed separately for the historical and future simulation. The text is now updated to correctly reflect this: "This index corresponds to normalized anomalies of the projection for each regime $i$ relative to the mean projection in the respective climate period, and the normalization is done with the climatological standard deviation of the projection in the respective climate period" (L155). The normalization coefficient is indeed different between ERA-Interim, CESM hist, and CESM eoc. The respective climate period is used to compute the normalization coefficient to ensure the normalized fields agree with the statistical basis for the weather regime projection.

L165. What is the sensitivity to this threshold? i.e. what was the proportion of no-regime days with the original threshold of 1?

The sensitivity is rather weak in the small range of the required threshold modification. With a threshold of 1.0 we would have 32.4% of no-regime days in CESM hist, and with a threshold of 0.98 we get 30.9%, which is very close to the 30.8% in ERA-Interim.

L176-180, Figure 1. Even if this is a single model study, a quick comment and comparison of these results with others in literature would be needed here.

We don't know with what to compare here, because to the best of our knowledge, this is the only study so far that applied the Grams et al. (2017) regime classification to climate change simulations. Other studies investigating regime frequency changes used other methods, which makes a meaningful comparison difficult.

L233. The sign of the ratio may be also interesting to investigate (i.e. do dynamic and thermodynamic effects contribute in the same direction?).

We agree, and we first differentiated between positive and negative values of gamma. We then realized that this made the plots and discussion more complex, without yielding too much additional insight. Therefore, we decided to reduce complexity here and consider in this study only the modulus of gamma.

L242. The order of magnitude estimate is fine, but I would avoid a more quantitative estimate at this stage (e.g. 10 -> 9-11%), since the frequency change could in principle be much larger.

We prefer to leave this sentence as is. It should be clear to the reader that this is just an estimate, and we would like to explain what a relative frequency change of 0.1 means in our case where typical values of $f_i$ are about 10%.

L260. For a "very good WR classification …". I understand that you expect delta Phi for each regime to be small with respect to the typical regime anomaly of Phi. However, the estimate is quite arbitrary (in principle, the regime intensity could increase/decrease by a more significant fraction) and a low value of this ratio could be due to other causes, rather than to the quality of the regime classification.

We don't fully understand this comment. With a "very good classification" we mean "good" in the sense that it stratifies well between higher and lower values of Phi. In this sense, $\text{Phi*}_{\text{hist,i}}$ is a reasonable indicator for a "good classification". To avoid confusion, we added "For a very good WR classification in terms of the variable Phi, …" (L254).

L277. I think it would be more appropriate here to set a more objective threshold on the climate change signal, e.g. where the signal is significant with respect to internal variability (e.g. considering the historical variability in a 10-year random sample). What does the 30% threshold correspond to in terms of climate change signal?

The (subjective) threshold for masking the gamma plots should not be considered as something too important. See also the reply to a comment of the first reviewer on p. 3-4 of

this document. The only aim of this masking is to avoid that the reader starts interpreting large values of $\gamma_i$ in regions where the question whether frequency changes of regimes contribute to the climate change signal, is not relevant, simply because the signal is small.

Fig. 2. I suggest to use the green/brown colorbar for the precipitation differences, as done in other figures. Also, it would be nice to add a title with the respective seasons on top (just a suggestion).

Thank you, we added small titles, but we decided to not change the colorbar. While we understand the point of the reviewer, we think that having a different colorbar better highlights the fact that what is shown in Fig. 2c,d (total climate change signal in precipitation, red/blue colorbar) is what we would like to "explain" in this study (i.e., the starting point of the study), whereas all the other climate change panels in Figs. 2-7 (brown/green colorbar) are for a specific weather regime and have a different role in our study.

L286. Also, negative values are found in the Mediterranean region, it may be worth commenting on that.

We now mention the negative values in the Mediterranean (L286).

L335. "weaker WR-specific anomalies". I think it would be worth adding in a supplementary a figure with the P* fields (hist and eoc, and/or the delta P fields), to allow comparison of the relative magnitude before multiplying by additional factors. Fig. A.17 and A.19 of the doctoral thesis would be very useful here.

Also in response to a similar suggestion by the first reviewer, we added contours of $P^*_{hist,i}$ to the panels in the 3$^{rd}$ column of Figs. 4-7 and to Fig. 3c.

L345. "but is more variable within a specific WR". I don't think this is indicated by the precipitation anomaly composite.

Indeed, this was not shown in the paper, and we decided to delete this remark.

L435. I appreciate the observation regarding the fact that intensity changes could also contain a dynamical signal, for example in terms of the amplitude of the dynamical anomalies. Do you see a way to extract some more information out of this, i.e. to separate a general thermodynamic response to regime-specific features?

No, unfortunately, we don't see a clean way of separating thermodynamic and dynamic signals, also because they can be interrelated in diverse and complex ways. In our view, more research is needed to understand regime-specific intensity changes.

L440. "$\Delta\Phi_i$ is fairly uniform across WRs". This is not true for all regions, for example the central North-Atlantic and the Iberian peninsula.

We agree with the reviewer, our formulation was not precise. We changed it to "While, at least in some regions, $\Delta\Phi_i(P)$ is fairly uniform across WRs, …" (L452).

---

## Referee Report (RR1)

Second review of Fischer et al. (egusphere-2024-1253)

Despite answering in detail to most of my comments, the authors did not fully understand the main issues I reported. I respect the authors' original methodological choices, but I still think some of the requests made in the first round of revision are legitimate and can be tackled without modifying the general approach. This does not change my overall opinion of the paper, which I consider an excellent piece of work and worth of publication. Nevertheless, I would like these two points to be more carefully addressed in the final version.

- Significance. I do not agree that performing significance tests with ensemble data is not meaningful since "there is no limit in generating more ensemble members", nor that "even tiny differences would appear as significant": in general, we do not know which changes are significant and which are not before performing a statistical test. It could well be that most of the changes are indeed significant, but this can't be taken for granted without a proper calculation.
The test I proposed is quite simple: take the ensemble standard deviation of 10-year averages in the historical period and, if the future-hist difference at a specific point is larger than that, that is an indication that the change is significant. I would apply this to regime composites of the change in Figs. 4,5,6,7: is the future-hist change larger (in absolute value) than the historical standard deviation of 10-year averages? Of course, this is only one possible measure of significance, but alternative ways are also possible. I don't think that this repeats Yettella and Kay (2017), since they did not investigate regime-specific changes.

- Interpretation. Your claim is quite strong, namely that regime frequency changes do not matter for the understanding of the impacts of future circulation changes on seasonal precipitation. However, I think the results you show are tightly linked with some of your methodological choices, and this should be discussed more in depth. In particular, these two choices impact the value of gamma:
  • since you separately removed the climatology on the historical and future ensembles (also, the calculation of the normalization factor is done separately), part of the climate change signal on the circulation is removed, thus reducing the amplitude of regime frequency changes;
  • on the other side, you do not remove the mean climatological change in the precipitation fields, so that regime intensity changes contain that signal (most of which is independent from the regimes, as you observe).

As far as I understand, the first choice reduces the "frequency effect", while the second enhances the "intensity effect". This is not to question your methodological choices - which are legitimate and are always to some extent "subjective" -, but to better inform the community that it is actually quite complicated to disentangle the two effects, and the results you get depend - at least to some extent - on some of your assumptions.

---

## Author Response (AR2)

*Paper egusphere-2024-1253*

**How relevant are frequency changes of weather regimes for understanding climate change signals in surface precipitation in the North Atlantic-European sector? – a conceptual analysis with CESM1 large ensemble simulations**

by Luise J. Fischer, David N. Bresch, Dominik Büeler, Christian M. Grams, Robin Noyelle, Matthias Röthlisberger, and Heini Wernli

*Replies (2nd round)*

We are grateful to the editor and reviewers for their additional comments that helped us to further improve the manuscript. Below we provide a one-to-one response to the specific points raised by the editor and reviewers. The comments from the editor and reviewers are in black and our replies in blue. Line numbers refer to the revised version of our paper.

We also would like to apologize for the long time it took us to do the (seemingly) minor revisions. The reasons were (i) lack of understanding of the first and last authors of the request for statistical testing, (ii) technical difficulties with re-accessing some of the data, and (iii) the busy schedule of the first author (who is now working outside academia). For the statistical testing, we invited advice from Robin Noyelle, a colleague in our group, and we added Robin to the list of co-authors. He helped us understanding the issue with the statistical testing raised by the 2nd reviewer and the editor, and he performed the testing (see below) and therefore contributed essentially to this new version of the paper. All co-authors agree with adding Robin to the list of co-authors.

**Editor**

Both of the original reviewers have provided an assessment of the revised manuscript, and appreciate all the work the authors have put in. I tend to agree with reviewer 2 that the two outstanding issues they mention are worth addressing. For significance, I think there has been a slight misunderstanding, but the reviewer clarifies this well – it is not about boosting weak signals by increasing the ensemble size but rather evaluating the significance of any/all identified WR-specific precip signals.

We addressed this point by performing a bootstrapping test (as explained in detail in our reply to the 2nd comment of reviewer 2.

If the authors could complete these minor revisions (clarification requested by reviewer 1, plus adding the significance assessment to the relevant figures and a few lines of discussion of how/whether the methodological choices affect the conclusions), I would see the manuscript as ready for publication. I plan to offer reviewer 2 the chance to see the manuscript one last time, but only if they have the time.

A few additional comments that may be helpful for the authors in finalizing the manuscript:

L59: multi-daily → multi-day or daily?

Changed to "multi-day".

L70: Suggested edit: Distinguishing between thermodynamic and dynamic climate change effects not only improves…, it also helps…

Thank you, changed as suggested.

L73: I'd remove "among others", or if absolutely necessary to keep it, move it to the end of the sentence.

Removed as suggested.

L79: "more positive phases" doesn't necessarily/exactly mean an increased frequency of NAO+.

Changed to "an increase in the frequency of the positive NAO phase".

General comments, applicable throughout manuscript:
- historic → historical. Thank you, changed in several places.

- incorrect usage of "respectively" (e.g., Fig. 2 caption). Deleted "respectively" in the caption of Fig. 2.

- a different colour scale for $\gamma_{overall}$ might help show differences better. The main intention is to show that values for $\gamma_{overall}$ are small, and this comes out best if we use the same colour scale as for the other $\gamma_i$ plots.

**Reviewer 1**

The authors have addressed my comments, and I have no further comment than the one below.

Caption of Figure 2: "Units days per day" Both the sentences and the units of the shading in panels (e) and (f) are not clear. The caption says number of wet days in which case the unit is days. However, the text in line 300 says that is a wet day frequency in which case the unit should be a % or a ratio between 0 and 1. Could the authors clarify the unit and text (caption or text)?

Thanks for spotting these inconsistencies. What we show is wet day frequency with values from 0 to 1. We clarify this in the captions and text.

**Reviewer 2**

Despite answering in detail to most of my comments, the authors did not fully understand the main issues I reported. I respect the authors' original methodological choices, but I still think some of the requests made in the first round of revision are legitimate and can be tackled without modifying the general approach. This does not change my overall opinion of the paper, which I consider an excellent piece of work and worth of publication. Nevertheless, I would like these two points to be more carefully addressed in the final version.

1) Significance. I do not agree that performing significance tests with ensemble data is not meaningful since "there is no limit in generating more ensemble members", nor that "even tiny differences would appear as significant": in general, we do not know which changes are significant and which are not before performing a statistical test. It could well be that most of the changes are indeed significant, but this can't be taken for granted without a proper calculation. The test I proposed is quite simple: take the ensemble standard deviation of 10-year averages in the historical period and, if the future-hist difference at a specific point is larger than that, that is an indication that the change is significant. I would apply this to regime composites of the change in Figs. 4–7: is the future-hist change larger (in absolute value) than the historical standard deviation of 10-year averages? Of course, this is only one possible measure of significance, but alternative ways are also possible. I don't think that this repeats Yettella and Kay (2017), since they did not investigate regime-specific changes.

We first struggled to develop an actionable plan to address this concern. Thanks to the expertise of Robin Noyelle, who joined the author team, we found a way to implement a bootstrapping approach to assess the statistical uncertainty in estimating the different terms induced by the finite size of the large ensemble. Our approach works as follows and this explanation is included in the revised paper at the end of Sect. 3: "In order to test the sampling sensitivity of the decomposition of $\Delta\varphi$ into WR-specific values to the 35 members used in each period, we implement a bootstrap procedure. For a given season and parameter $\varphi$ we resample randomly with replacement over all members the same numbers of days as in the original data set. We then compute the $\Delta\varphi_i$, the decomposition (Eq. 5) and the $\gamma_i(\varphi)$ parameter in this resampled data set for each WR at each grid point. We repeat this procedure 1000 times in order to give an estimation of the sampling distribution of the different terms. We then compute the probability in this resampled distribution that the value 0 (for $\Delta\varphi_i$ and the terms in the decomposition of Eq. 5) or 1 (for $\gamma_i(\varphi)$) is exceeded or subceeded. This gives a bootstrapped p-value for significance at each grid point. We then use a false discovery rate test of 0.1 to take into account spatial correlations (Wilks, 2016) and flagged as significant the grid points that pass this test."

2) Interpretation. Your claim is quite strong, namely that regime frequency changes do not matter for the understanding of the impacts of future circulation changes on seasonal precipitation. However, I think the results you show are tightly linked with some of your methodological choices, and this should be discussed more in depth. In particular, these two choices impact the value of gamma:

- since you separately removed the climatology on the historical and future ensembles (also, the calculation of the normalization factor is done separately), part of the climate change signal on the circulation is removed, thus reducing the amplitude of regime frequency changes.

  We understand your point of view but still think that there is no other straightforward way to address regime frequency changes between different climates. We see the value of weather regimes in describing the variability of the flow relative to the respective climatological mean, and therefore we separately remove the climatology in both climate periods.

- on the other side, you do not remove the mean climatological change in the precipitation fields, so that regime intensity changes contain that signal (most of which is independent from the regimes, as you observe).

  This is because we would like to "understand", i.e., decompose the climate change signal in precipitation. We explained in the last replies that removing the mean climatological change in precipitation would lead to a different study addressing a different research question.

As far as I understand, the first choice reduces the "frequency effect", while the second enhances the "intensity effect". This is not to question your methodological choices – which are legitimate and are always to some extent "subjective" – but to better inform the community that it is actually quite complicated to disentangle the two effects, and the results you get depend – at least to some extent – on some of your assumptions.

We are grateful for your critical perspective on our decomposition, which made us think again about what we regard as the most appropriate way of addressing the main question of our study (as phrased in the title of the paper). Without someone implementing your alternative approach (removing climate change signal in precipitation and calculating the regimes in the future climate not relative to the future climatology), we cannot fully clarify the pros and cons of our conceptually different viewpoints. We therefore add a brief remark in the conclusions as follows: "During the review process the concern was brought up by one of the reviewers that the values of $\gamma_i(\varphi)$ are relatively small because we separately removed the climatology on the historical and future ensembles and therefore removed part of the climate change signal on the circulation, thus reducing the amplitude of regime frequency changes. However, since we regard weather regimes as patterns describing the circulation variability relative to the respective climatological mean, we don't see how regime frequency changes between different climates can be calculated in a way that is different from the one we implemented. We trust that future studies can further elaborate on these important conceptual questions."